# LEARNING WITH TEMPORAL LABEL NOISE

## ABSTRACT

Many sequential classification tasks are affected by *label noise* that varies over time. Such noise might arise from label quality improving, worsening, or periodically changing over time. In this work, we formalize the problem of label noise in sequential classification, where the labels are corrupted by a temporal, or time-dependent, noise function. We call this novel problem setting *temporal label noise* and develop a method to learn a sequential classifier that is robust to such noise. Our method can estimate the temporal label noise function directly from data, without *a priori* knowledge of the noise function. We first demonstrate the importance of modelling the temporal label noise function and how existing methods will consistently underperform. In experiments on both synthetic and real-world sequential classification tasks, we show that our algorithm leads to state-of-the-art performance in the presence of diverse temporal label noise functions.

## 1 INTRODUCTION

Many supervised learning datasets contain *noisy* observations of ground truth labels. Such *label noise* can arise due to issues in annotation or data collection, including lack of expertise in human annotation [22, 51], discrepancies in labelling difficulty [11, 21, 51], subjective labeling tasks [34, 39, 45], and systematic issues in automatic annotation like measurement error [23, 36].

Label noise is a key vulnerability of modern supervised learning [14]. Intuitively, when training a model on data with noisy labels, the model can learn to predict noise instead of true signals. During testing, the model will drastically underperform on tasks that require accurately predicting the ground-truth. This problem is exacerbated by deep learning methods, which have the representational capacity to memorize all the noise in a dataset, leading to poor model performance [1, 12, 13, 26].

But existing label noise methods are all built for *static*, time-invariant data. In reality, practical problems often involve time series data, which involve labels collected over time. Here, we pose that label noise can follow *temporal* trends. And by accounting for such temporal label noise, we will achieve more robust models. To the best of our knowledge, learning from temporal label noise is unstudied, yet clearly exists in a range of tasks. For example:

*Human Activity Recognition*. Wearable device studies commonly ask participants to report their activities over time. In this case, annotators may mislabel certain periods due to factors such as recall bias or labelling at random to achieve a participation incentive [20, 44].

*Longitudinal Self-Reporting for Mental Health*: Mental health studies often collect self-reported survey data over long periods of time. Such self-reporting is known to be biased [5, 37, 38, 48], where participants may be more or less likely to report certain features. For example, the accuracy of self-reported alcohol consumption is known to be seasonal [7].

*Clinical Measurement Error*: In machine learning for health, the labels used for learning come from healthcare provider notes in an electronic health record. These providers may produce noisier annotations in electronic health records during busier times or when a patient is deteriorating or the bedside situation is more chaotic [50].

Classifying time series in the presence of temporal label noise is challenging. First, existing losses that are robust to static label noise under-perform when facing temporal noise—as we show in our experiments. This means that classifiers can still suffer, even when the temporal noise function is known. Worse yet, the temporal label noise function is generally *not* known. Often times, all that is

**Figure 1:** a) Temporal noise processes degrade label quality in sequential labeling tasks. b) Our method directly models any temporal label function. Modeling this function allows us to train strong models in the face of noise.

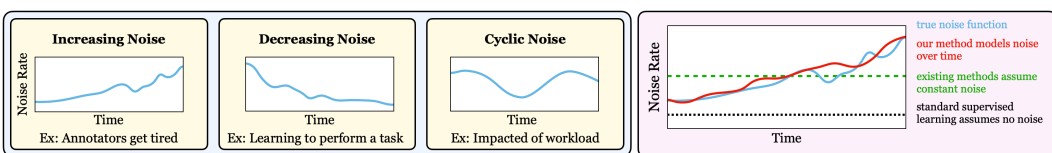

a) Temporal noise functions degrade label quality in sequential annotation

b) Our method directly models any temporal noise function

available is a dataset with temporally noisy labels with no indication of which instances or time-steps are more likely to be correct.

In this work, we propose two temporal loss functions that we prove are *robust to temporal label noise* in the case where the temporal noise function is known. Motivated by this, we propose a novel time series classification objective, TENOR, which *learns* the temporal noise function from the data, while simultaneously leveraging our noise-robust loss. TENOR allows us to train accurate, noise-tolerant time-series classifiers even in the case of unknown, temporal label noise.

Our contributions are as follows:

1. We formalize the problem of learning from noisy labels in temporal settings. We are the first to study this important problem on which prior works under-perform.

2. We propose two loss functions for training models that are robust to temporal label noise. On their own, each can be used to improve prior methods.

3. We propose TENOR, the first method for explicitly learning temporal noise functions. By pairing a neural network with our proposed loss functions, TENOR can model any noise function and thereby lead to better classifiers.

4. We perform an extensive empirical study on real data, and find that existing methods under-perform in the presence of temporal noise. Meanwhile, our proposed methods lead to better classifiers, highlighting the importance of accounting for temporal noise to achieve performant models.

## 2 RELATED WORK

Learning with noisy labels is a well-studied area of research [3, 28, 30, 46]. Many approaches utilize heuristics for identifying correctly-labeled instances [18, 55], or regularization techniques to minimize the impact of incorrect labels [19, 26].

Our work contributes to *noise-robust loss functions* [16, 25, 28, 33, 49], as these approaches allow for leveraging arbitrarily complex model architectures while still retaining theoretical guarantees. Consequential work in this area has shown that noise-robust loss functions can be created with accurate knowledge of the underlying noise process [33, 35]. These methods have strong statistical guarantees. In reality, the label noise function is not known *a priori*. Recent work has contributed techniques to estimate this function from noisy data alone [24, 25, 35, 52, 54, 56]. These methods make certain assumptions, as the noise function is generally unidentifiable otherwise [27, 52]. For example. the *anchor point assumption* assumes that there are some extreme-probability instances that determine the values of the transition matrix [35]. A softer assumption is made by Li et al. [25], who assume instead that the clean posterior is sufficiently scattered [15]. However, unlike our work, *they do not consider temporally evolving noise functions.*

Nearly all of these existing works have focused on the static noise setting, where the label noise does not change over time. While a few methods have considered noise in time series data [2, 10], they focus on heuristics rather than developing temporal-noise-robust methods. Crucially, these works still do not consider temporally evolving noise.

## 3   PROBLEM STATEMENT

In this section, we introduce and formalize the new problem of learning with noisy labels under temporal noise for sequential classification tasks.

We consider a data generating function with random variables $(\mathbf{x}_{1:T}, \mathbf{y}_{1:T}, \tilde{\mathbf{y}}_{1:T}) \sim P_{\mathbf{x}_{1:T}, \mathbf{y}_{1:T}, \tilde{\mathbf{y}}_{1:T}}$, where $\mathbf{x}_{1:T} \in \mathcal{X} \subset \mathbb{R}^{d \times T}$ is a sequence of $T$ feature vectors, $\mathbf{y}_{1:T} \in \mathcal{Y} = \{1, \ldots, C\}^T$ is a sequence of $T$ *clean* labels, and $\tilde{\mathbf{y}}_{1:T} \in \mathcal{Y} = \{1, \ldots, C\}^T$ is a sequence of $T$ *noisy* labels. When learning from temporal label noise, we do not observe the *true* label sequence $\mathbf{y}_{1:T}$. Thus, we only have access to a set of $n$ instances $D = \{(\boldsymbol{x}_{1:T}, \tilde{\boldsymbol{y}}_{1:T})_i\}_{i=1}^n$.

We make a standard conditional independence assumption [4, 9] that a label $\mathbf{y}_t$ at time $t$ only depends on the prior history of feature vectors $\mathbf{x}_{1:t}$:

$$p(\mathbf{y}_{1:T} \mid \mathbf{x}_{1:T}) = \prod_{t=1}^{T} p(\mathbf{y}_t | \mathbf{x}_{1:t}). \tag{1}$$

In addition, we assume the sequence of noisy labels is independent of the features given the true labels: $\tilde{\mathbf{y}}_{1:t} \perp\!\!\!\perp \mathbf{x}_{1:t} \mid \mathbf{y}_{1:t}$ for all $t = 1, \ldots, T$. Thus, we can express the conditional probability of observing noisy labels at time $t$ as:

$$q_t(\tilde{\mathbf{y}}_t | \mathbf{x}_{1:t}) = \sum_{c=1}^{C} q_t(\tilde{\mathbf{y}}_t \mid \mathbf{y}_t = c) p(\mathbf{y}_t = c \mid \mathbf{x}_{1:t}), \tag{2}$$

where $q_t$ is a *time-dependent* quantity that captures the probability that $\tilde{\mathbf{y}}_t \neq \mathbf{y}_t$ across time-steps $t$. Together, the assumptions in Eq. (1) and Eq. (2) imply that $p(\tilde{\mathbf{y}}_{1:T} | \mathbf{x}_{1:T}) = \prod_{t=1}^{T} q_t(\tilde{\mathbf{y}}_t | \mathbf{x}_{1:t})$.

Our goal is to use the dataset of *noisy* labels to learn a classification model $\boldsymbol{h}_\theta : \mathcal{X} \to \Delta^{C-1}$ that maximizes accuracy under the *clean* labels. We assume that the model has parameters $\theta$ where:

$$\theta \in \underset{\theta}{\operatorname{argmax}} \, \mathbb{E}_{\mathbf{y}_{1:T} | \mathbf{x}_{1:T}} \big[ p(\mathbf{y}_{1:T} = \boldsymbol{h}_\theta(\mathbf{x}_{1:T}) \mid \mathbf{x}_{1:T}) \big]$$

Let $\boldsymbol{h}_\theta(\boldsymbol{x}_{1:t}) = \boldsymbol{\psi}^{-1}(\boldsymbol{g}_\theta(\boldsymbol{x}_{1:t})) : \mathbb{R}^{d \times t} \to \Delta^{C-1}$ be a sequence classifier model, such that $\boldsymbol{g}_\theta(\boldsymbol{x}_{1:t})) :\to \mathbb{R}^{d \times t} \to \mathbb{R}^C$ is the pre-activation of output of the model and $\boldsymbol{\psi} : \Delta^{C-1} \to \mathbb{R}^C$ is an invertible link function that maps the model's output to a probability distribution [40]. For example, when $h_\theta$ is a neural network classifier, $g_\theta$ is the final logits and $\boldsymbol{\psi}^{-1}$ is the softmax function.

## 4   LEARNING FROM NOISY LABELS IN TIME SERIES

In this section, we start by introducing a way to model temporal label noise, using a Temporal Label Noise Function. We then prove that knowledge of this Temporal Label Noise Function allows for the training of a sequential classifier $\boldsymbol{h}_\theta$ that can predict the *clean* labels given samples from a dataset $D$ with noisy labels. Finally, motivated by these proofs, we propose methods for effectively learning temporal label noise functions from datasets with noisy labels.

### 4.1   MODELLING TEMPORAL LABEL NOISE

We encode the conditional probabilities of observing a noisy label at time $t$ in matrix $\boldsymbol{Q}_t = [q_t(\tilde{\mathbf{y}}_t = i \mid \mathbf{y}_t = j)]_{i,j} \in \mathbb{R}^{C \times C}$, and assume that $Q_t$ is the output of a *Temporal Label Noise Function* at time $t$:[1]

**Definition 1.** The *temporal label noise function* is a matrix-valued function $\boldsymbol{Q} : \mathbb{R}_+ \to [0,1]^{c \times c}$ specifies the label noise distribution over time. For any given time $t > 0$, $\boldsymbol{Q}(t)$ is a $c \times c$ positive, row-stochastic, and diagonally dominant matrix whose entries encode the probability of flipping the label observed at time $t$: $\boldsymbol{Q}(t)_{j,k} := p(\tilde{\mathbf{y}}_t = k \mid \mathbf{y}_t = j)$.

---

[1] We deviate from standard nomenclature in the noisy labels literature to avoid confusion with $t$ as time.

The temporal label noise function can capture a wide variety of temporal noise as shown in Fig. 1. For example, generating a smooth function $\boldsymbol{Q}(t)$ that represents cyclic noise simply requires generating a $c \times c$ matrix where each entry of $\boldsymbol{Q}(t)$ is parameterized by a periodic function $f : \mathbb{R}_+ \to [0, 1]$ and the output matrices are row-stochastic. Another example is a scenario where a human annotator generating labels learns to perform the task better over time. In this case, we expect label noise to reduce over time, and entries of $\boldsymbol{Q}(t)$ can be parameterized by an exponential decay function.

## 4.2 LOSS CORRECTION FOR TEMPORAL LABEL NOISE

In what follows, we introduce loss functions that we prove are tolerant to temporal label noise when the true noise function is known. The proofs of our theoretical claims are given in Appendix A. To begin, we first define the *backward sequence loss* , which extends the backward loss of Patrini et al. [35] to a temporal setting.

**Definition 2.** The *backward sequence loss* is the function:

$$\overleftarrow{\ell}_{seq}(\mathbf{y}_{1:T}, \boldsymbol{h}_\theta(\mathbf{x}_{1:T})) := \sum_{t=1}^{T} \overleftarrow{\ell}_t(\mathbf{y}_t, \boldsymbol{h}_\theta(\mathbf{x}_{1:t}))$$

where $\overleftarrow{\ell}_t(c, \boldsymbol{h}_\theta(\boldsymbol{x}_{1:t})) = [\boldsymbol{Q}_t^{-1}]_{c,:} \cdot \boldsymbol{\ell}(\boldsymbol{h}_\theta(\boldsymbol{x}_{1:t}))$. Here, $\boldsymbol{\ell}(\boldsymbol{h}_\theta(\mathbf{x}_{1:t})) := [\ell(c, h_\theta(\mathbf{x}_{1:t}))]_{c=1:C}^\top$ is a vector containing the negative log-likelihood (NLL) loss of $\ell(c, h_\theta(\mathbf{x}_{1:t}))$ for observed class $c$.

Intuitively, the *backward sequence loss* removes noise from noisy labels by inverting the noise function prior to the *backward* pass of a deep learning algorithm [35]. An intriguing property of *backward sequence loss* is shown in Theorem 1.

**Theorem 1.** Minimizing the expected *backward sequence loss* over noisy label sequences is equivalent to maximizing the likelihood over clean label sequences.

$$\underset{\theta}{\operatorname{argmax}} \, \mathbb{E}_{\mathbf{y}_{1:T}|\mathbf{x}_{1:T}} \log \left( p_\theta(\mathbf{y}_{1:T}|\mathbf{x}_{1:T}) \right) = \underset{\theta}{\operatorname{argmin}} \, \mathbb{E}_{\tilde{\mathbf{y}}_{1:T}|\mathbf{x}_{1:T}} \overleftarrow{\ell}_{seq}(\boldsymbol{h}_\theta(\mathbf{x}_{1:T}))$$

This means that we can *train on the noisy distribution* and still learn an *optimal classifier for the clean distribution*.

A separate strategy is to consider the noisy posterior as the matrix-vector product of a noise transition function and a clean posterior (Eq. (2)). To this effect, we also define the *forward sequence loss* :[2]

**Definition 3.** The *forward sequence loss* is the function:

$$\overrightarrow{\ell}_{seq,\psi}(\boldsymbol{y}_{1:T}, \boldsymbol{g}_\theta(\boldsymbol{x}_{1:T})) = \sum_{t=1}^{T} \overrightarrow{\ell}_{t,\psi}(y_t, \boldsymbol{g}_\theta(\boldsymbol{x}_{1:t}))$$

where $\overrightarrow{\ell}_{t,\psi}(y_t, \boldsymbol{g}_\theta(\boldsymbol{x}_{1:t})) := \ell_t(y_t, \boldsymbol{Q}_t^\top \cdot \boldsymbol{\psi}^{-1}(\boldsymbol{g}_\theta))$ is a proper composite loss.

The *proper composite loss* statement refers to the fact that $\overrightarrow{\ell}_{t,\psi}(\cdot, \cdot)$ is a loss that is well-calibrated for probability estimation (e.g., NLL loss) that incorporates the link function $\psi$ to map model outputs into probability estimates (see [40] for more detail).

We show that *forward sequence loss* is a *robust* loss:

**Theorem 2.** A classifier that minimizes the empirical *forward sequence loss* over the noisy labels maximizes the empirical likelihood of the data over the clean labels.

$$\underset{\theta}{\operatorname{argmin}} \, \mathbb{E}_{\tilde{\mathbf{y}}_{1:T}, \mathbf{x}_{1:T}} \overrightarrow{\ell}_{seq,\psi}(\mathbf{y}_{1:T}, \boldsymbol{g}_\theta(\mathbf{x}_{1:T})) = \underset{\theta}{\operatorname{argmin}} \sum_{t=1}^{T} \mathbb{E}_{\mathbf{y}_{1:t}, \mathbf{x}_{1:t}} \ell_{t,\phi}(\mathbf{y}_{1:T}, \boldsymbol{g}_\theta(\mathbf{x}_{1:T})).$$

---

[2]Note that the *backward sequence loss* and *forward sequence loss* differ from the Forward-Backward algorithm used in Hidden Markov Model inference.

The *forward sequence loss* can be intuitively understood as decomposing the noisy posterior distribution into a product of a noise function and a clean posterior distribution. This operation takes place during the *forward* pass of a deep learning algorithm.

Theorems 1 and 2 show that the sequential loss functions can learn accurate classifiers from sequences of noisy labels when we know the temporal noise function $\boldsymbol{Q}(t)$. Given that $\boldsymbol{Q}(t)$ is not known in most practical settings, methods should be able to estimate $\boldsymbol{Q}(t)$ without *a priori* knowledge of the noise.

### 4.3 LEARNING THE TEMPORAL LABEL NOISE FUNCTION

It is well-known that noise transition matrices can only be identified under certain assumptions. One general assumption proposed in recent work by Li et al. [25] is to assume that $\boldsymbol{Q}$ corresponds to a *minimum-volume simplex* that contains the posterior distribution, $\boldsymbol{p}(\mathrm{y} \mid \mathbf{x})$. In a static classification task, this assumption implies that $\boldsymbol{p}(\tilde{\mathrm{y}} \mid \mathbf{x}) = \boldsymbol{Q}^\top \boldsymbol{p}(\mathrm{y} \mid \mathbf{x})$ – i.e., that the columns of $\boldsymbol{Q}^\top$ encloses the noisy posterior for any $\boldsymbol{x}$. Fu et al. [15] prove that a sufficient condition for $\boldsymbol{Q}$ to be identifiable in this setting is if the posterior distribution is *sufficiently scattered* over the unit simplex (for details see [15, 25]).

We propose a technique to model and learn the temporal label noise function $\boldsymbol{Q}(t)$ under the minimum volume simplex assumption [25] by minimizing the volume of $\hat{\boldsymbol{Q}}_t$ across time. Specifically, we estimate the label noise matrix $\boldsymbol{Q}(t)$ for each time $t = 1, \ldots, T$ such that:

$$\hat{\boldsymbol{Q}}_t \in \arg\min_{\boldsymbol{Q}} \ \mathrm{vol}(\boldsymbol{Q}) \quad \text{s.t.} \quad \hat{\boldsymbol{Q}}_t^\top h_\theta(\boldsymbol{x}_{1:t}) = p(\tilde{y}_t \mid \boldsymbol{x}_{1:t}) \text{ for } t = 1, \ldots, T \qquad (3)$$

We assume that the $\hat{\boldsymbol{Q}}_t$ for each time $t$ is generated by a temporal label noise function as defined in Definition 1. We learn this function from data using a fully connected neural network with parameters $\omega, \boldsymbol{Q}_\omega(\cdot) : \mathbb{R} \to [0,1]^{c \times c}$. With clever implementation, we can ensure that the outputs of this network meet Definition 1 (see Appendix D.2).

In order to solve the equality-constrained optimization problem above, we employ the augmented Lagrangian method [6]. Considering the equality-constraint is achieved when the loss is 0, we can denote $R_t(\theta, \omega) = \frac{1}{n} \sum_{i=1}^n \ell_t(y_{t,i}, \boldsymbol{Q}_\omega(t)^\top h_\theta(\boldsymbol{x}_{1:t,i}))$ which re-formulates the constrained objective into the following unconstrained objective:

$$\mathcal{L}(\theta, \omega) = \frac{1}{T} \left[ \sum_{t=1}^T \|\boldsymbol{Q}_\omega(t)\|_F + \lambda \sum_{t=1}^T R_t(\theta, \omega) + \frac{c}{2} |\sum_{t=1}^T R_t(\theta, \omega)|^2 \right]$$

where $\lambda \in \mathbb{R}$ is the Lagrange multiplier and $c > 0$ is the penalty parameter. Given the constraints imposed in Definition 1, minimizing the *Frobenius norm* of $\boldsymbol{Q}$, a convex function, amounts to minimizing the volume of $\boldsymbol{Q}$. The augmented Lagrangian method gradually increases the penalty parameter until the constraint is eventually met, and $\lambda$ converges to the Lagrangian multiplier of Eq. (3) [6]. Further details of its implementation can be found in Appendix B.

We refer to this as the **Te**mporal **No**ise **R**obust (TENOR) objective. The TENOR objective assumes that label noise arises from a single matrix-valued function with parameters $\omega$, which couples the estimates of $\boldsymbol{Q}_t$ across time periods $t = 1, \ldots, T$.

### 4.4 TIME-SEPARABLE ESTIMATION

We pair our methods with two methods that estimate $\boldsymbol{Q}(t)$ at each time $t$. Each method makes different assumptions surrounding the temporal noise function and its identifiability. The methods may be useful in settings where these assumptions are plausible. For example, if $Q_\omega$ does not have sufficient representational capacity to model the underlying noise function, it might be prudent to estimate $\boldsymbol{Q}(t)$ independently at each timestep to avoid enforcing continuity between time steps. Lastly, these methods serve as competitive baselines for our method described in Section 4.3, as no existing $\boldsymbol{Q}$-estimation technique is built to handle temporal noise. These techniques broadly show how any static estimation technique can be extended to the temporal setting.

**AnchorTime** One assumption to ensure identifiability of $\boldsymbol{Q}$ is that a dataset contains *anchor points* whose labels are known to be correct [27, 35, 52]. In a sequential setting, anchor points correspond to instances that maximize the probability of belonging to a particular class at a particular time step:

$$\bar{\boldsymbol{x}}_t^j = \arg\max_{\boldsymbol{x}_t} p(\tilde{y}_t = j \mid \boldsymbol{x}_{1:t}) \tag{4}$$

**Assumption 1** (Existence of Anchor Points). *For each class $j$ and time $t$, there exists a non-empty subset $\bar{\mathcal{X}}_t^j \subset \mathcal{X}$ such that $\bar{\mathcal{X}}_t^j := \{\bar{\mathbf{x}}_t^c \mid p(y_t = j \mid \bar{\mathbf{x}}_{1:t}) = 1\}$.*

Since $p(y_t = j \mid \bar{\mathbf{x}}_{1:t}^j) = 1$ for the clean label, we can express each entry of the label noise matrix as:

$$\hat{\boldsymbol{Q}}(t)_{j,k} = p(\tilde{y}_t = k \mid \bar{\boldsymbol{x}}_{1:t}^j). \tag{5}$$

We can construct $\hat{\boldsymbol{Q}}(t)$ using a two-step approach that is analogous to the approach of Patrini et al. [35] for static prediction tasks:

1. Fit a probabilistic classifier to predict noisy labels from the observed data.

2. For each class $j \in \mathcal{Y}$ and time $t \in [1 \ldots T]$:

    i Identify anchor points for class $j$: $\bar{\boldsymbol{x}}_t^j = \arg\max_{\boldsymbol{x}_t} p(\tilde{y}_t = j \mid \boldsymbol{x}_{1:t})$.

    ii Set $\hat{\boldsymbol{Q}}(t)_{j,k}$ as the probability of classifier predicting class $j$ at time $t$ given $\bar{\boldsymbol{x}}_t^j$.

**VolMinTime** An alternative strategy to Section 4.3, is to assume that there is no temporal relationship between $\boldsymbol{Q}_t$ across time, and treat each time step independently. Using the *minimum-volume simplex* described in Section 4.3, we can learn the model and the label noise matrix by simultaneously optimizing the empirical *forward sequence loss* on noisy labels and the aggregate volume of $\hat{\boldsymbol{Q}}_t$ over $t \in [1 \ldots T]$:

$$\mathcal{L}(\theta, \hat{\boldsymbol{Q}}_1, \ldots, \hat{\boldsymbol{Q}}_T) = \frac{1}{n} \sum_{i=1}^n \sum_{t=1}^T \ell_t(y_{t,i}, \hat{\boldsymbol{Q}}_t^\top h_\theta(\boldsymbol{x}_{1:t,i})) + \lambda \cdot \frac{1}{T} \sum_{t=1}^T \log \det(\hat{\boldsymbol{Q}}_t) \tag{6}$$

The objective in Eq. (6) minimizes the volume of $\boldsymbol{Q}$ using the $\log \det$ of a square matrix [25].

In Eq. (6), each $\hat{\boldsymbol{Q}}_t$ is parameterized with a separate set of trainable real-valued weights, which are learned independently with the data from time $t$ using a standard convex optimization algorithm (e.g., gradient descent). This provides a direct time-series modification of a state-of-the-art technique for noise transition matrix estimation in the static setting [25]. Intuitively, this method allows for $C \times C \times T$ weights representing the noise function at each time step to be learned independently, as opposed to TENOR which explicitly learns a parametric representation of the noise function.

## 5 Experiments

We evaluate our approach on synthetic and real-world classification tasks.[3] We compare to state-of-the-art approaches, using both their original formulation, and the augmented version for the temporal setting that we derive in Section 4.2. Additional details and results are provided in the Appendix D.

### 5.1 Experimental Setup

**Datasets** We use one synthetic (`synth`) and four real-world classification datasets (binary and multiclass), consisting of sequential accelerometer data for human activity recognition (`har` [41], `har70` [29]) and continuous EEG signals for sleep detection (`eeg_sleep` [17]) and blink detection (`eeg_eye` [42]). To validate our setup in an ideal setting, we generate our synthetic dataset according to the exact data-generating assumptions in Eq. (1). By using synthetic data, we can be sure that variance in model performance is due to characteristics of the data, not challenges of learning on real data. For all datasets, in order to evaluate each model's robustness to label noise, we consider labels in the training data to be "ground-truth" clean labels and flip the labels using one of six temporal noise functions. These functions are: *Time Independent*, *Exponential* decay, *Linear* decay, *Sigmoid*, *Sinusoidal*, and *Mixed* (Exponential for one class and Sigmoid for the other). Further details and visualizations of these noise functions are available in Appendix D.

---

[3] We include code to reproduce our results in an anonymized repository.

**Table 1:** Comparison of clean test set accuracy (%) and $\hat{Q}(t)$ reconstruction error (MAE) across all datasets and methods. We show results for static methods at the top and their temporal variants at the bottom. The noise function is *Mixed* and fixed to 30%. ± is the st. dev. over 10 runs.

|  |  | synth | | har | | har70 | | eeg_eye | | eeg_sleep | |
|---|---|---|---|---|---|---|---|---|---|---|---|
|  |  | Accuracy ↑ | MAE ↓ | Accuracy ↑ | MAE ↓ | Accuracy ↑ | MAE ↓ | Accuracy ↑ | MAE ↓ | Accuracy ↑ | MAE ↓ |
| Static | Uncorrected | 79.2±1.2 | – | 70.6±3.0 | – | 77.3±1.7 | – | 71.3±.8 | – | 68.2±2.0 | – |
| | Anchor | 80.8±0.7 | .16±.00 | 79.1±2.6 | .13±.01 | 79.3±.01 | .11±.01 | 75.1±1.1 | .10±.01 | 65.9±2.3 | .11±.00 |
| | VolMinNet | 85.6±0.9 | .10±.00 | 85.1±2.7 | .11±.00 | 81.0±0.7 | .11±.00 | 73.2±1.4 | .13±.00 | 70.3±2.2 | .11±.00 |
| Temporal | AnchorTime | 85.3±0.8 | .1±.01 | 80.0±1.8 | .11±.01 | 81.2±1.1 | .08±.00 | 79.5±1.8 | .06±.01 | **70.4±2.8** | .06±.00 |
| | VolMinTime | 85.8±0.5 | .08±.00 | 82.4±2.7 | .10±.00 | 82.0±1.2 | .08±.00 | 76.9±1.8 | .08±.00 | 70.3±3.9 | .09±.01 |
| | TENOR | **94.4±1.0** | **.03±.01** | **95.8±2.2** | **.03±.01** | **89.0±0.3** | **.02±.00** | **83.7±0.5** | **.01±.00** | 70.4±2.3 | **.04±.01** |

**Figure 2:** Comparing performance of models trained with *backward sequence loss* and *forward sequence loss* on synth with varying degrees of temporal label noise using either the true temporal noise function (Temporal) or the average temporal noise function (Static). Error bars are st. dev. over 10 runs.

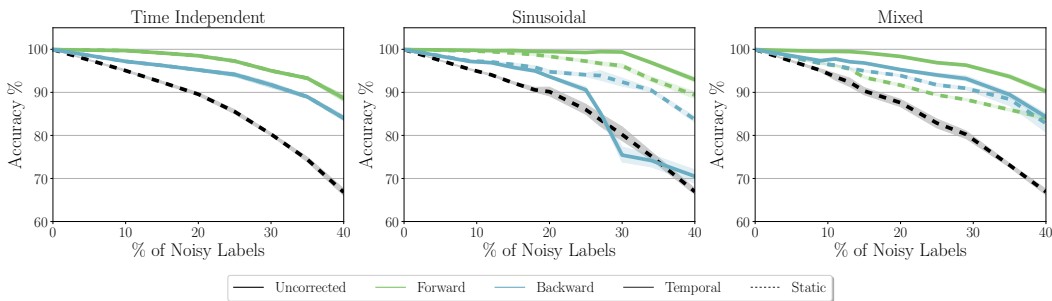

**Compared Methods** We train a sequential classifier for each dataset and compare TENOR to five alternatives from two groups: 1) **Static.** We consider an Uncorrected learning baseline (NLL loss), which assumes no noise exists and two *static* methods: Anchor [27, 35, 52] and VolMinNet [25]. 2) **Temporal.** We extend the static methods into our temporal setting, as described in Section 4.4. We refer to these as AnchorTime, VolMinTime, respectively.[4]

**Implementation Details** We use a Recurrent Neural Network with *Gated Recurrent Units* (GRU) [8] as our classifier for all experiments, since it can easily predict one class label per time step. TENOR uses an additional fully-connected neural network with 10 hidden layers to estimate $Q(t)$. More details on how these networks are constructed can be found in Appendix D.2.

**Evaluation** We split each dataset into a *noisy* training sample (80%) and a *clean* test sample (20%). We measure model accuracy using *clean test-set accuracy* and we also evaluate how well each method learns the underlying temporal noise function, using the Mean Absolute Error (MAE) between the true $Q_t$ and estimated $\hat{Q}_t$ for all $t$.

## 5.2 RESULTS AND DISCUSSION

In this section we show through various experiments the importance of modelling the temporal noise for learning models that are more robust to noise present in labels.

**Modeling Temporal Noise Improves Classification Performance** First, we show clear value in accounting for temporal label noise. Table 1 shows the performance of each method on all five datasets. We find that overall, the temporal methods are consistently more accurate than their non-temporal counterparts highlighting the importance of modelling temporal noise in these settings. Among the temporal methods, TENOR achieves best performance in comparison to AnchorTime

---

[4]VolMinTime can also be trained using the Frobenius norm of $Q$ to minimize the volume, and it can also be optimized with the augmented Lagrangian method. We include these experiments in Appendix E.5 but find they make no difference empirically.

**Figure 3:** Comparison of clean test set Accuracy (%) for `har` across varying degrees of temporal label noise comparing Uncorrected Loss, *Static* Methods (Top Row), *Temporal* Methods (Bottom Row), and TENOR. Error bars are st. dev. over 10 runs.

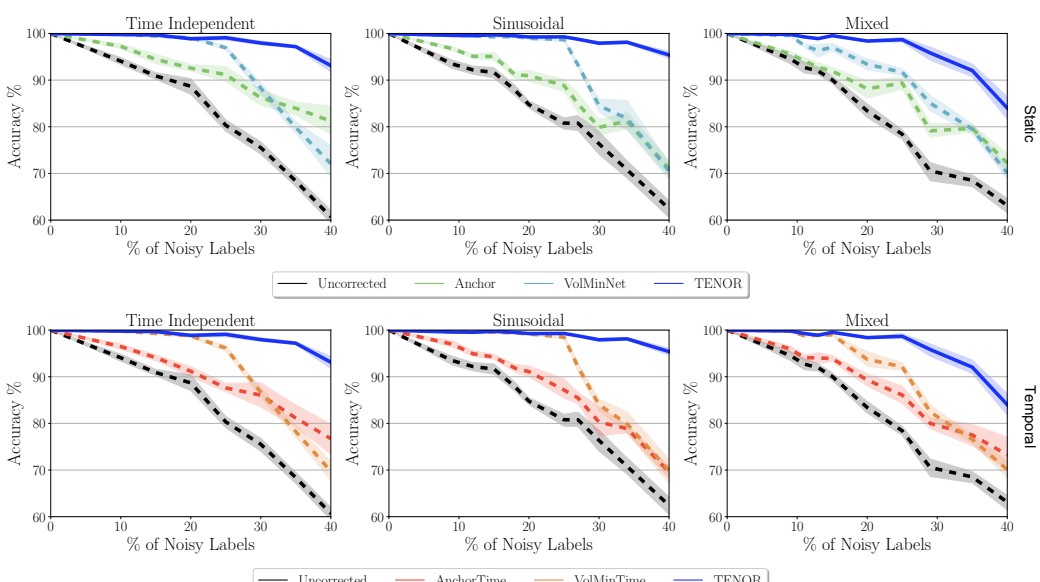

and VolMinTime. TENOR's superiority is even clearer when compared to the original non-temporal versions of these methods. We also notice that TENOR achieves the best reconstruction error on all datasets in terms of MAE. This is likely evidence for a direct correlation between learning the noise function and classification accuracy on real data.

As demonstrated in Fig. 3, these benefits hold across different degrees of label noise and across different functional forms of label noise. More importantly, the benefit of TENOR becomes more evident as the amount of noise increases in the data. In all these cases, we observe that the temporal methods are consistently more robust to both temporal and static label noise. We show in Fig. 2, in situations where the temporal noise function is truly uniform in time, both approaches yield the same performance – suggesting that there is nothing to lose by modeling temporal effects.

Overall, these findings suggest that we can improve performance by explicitly modeling how noise varies across time – i.e., this performs better than assuming it is distributed uniformly in time.

**Ignoring Temporal Noise Leads to Poor Performance**  We next demonstrate the limitations of *static* approaches. As our theoretical results in Section 4.2 show, we expect that accurate knowledge of the noise process can lead to noise-tolerant models. However, existing $Q$-estimators assume label noise is static. We validate this experimentally by comparing our *forward sequence loss* method using the true temporal noise function vs. a static approximation (average of noise over time). As shown in Fig. 2, static approximation consistently leads to poor performance. These differences are more pronounced in the Mixed noise function, suggesting that temporally-variable noise may exacerbate this weakness. We also verify that the differences vanish when the underlying temporal noise process is itself uniform—here, the static approximation and true temporal noise function are identical.

**TENOR successfully learns temporal noise functions**  Our results suggest that performance gains from our method are linked to how well we estimate the temporal label noise function. In Table 2, we show that our method consistently estimates the noise function with lower mean absolute error across different families of noise function. The results presented in this table are binary classification on the `har` dataset, but results on all other datasets (including multiclass) are present in Appendix E. Qualitatively, we can compare our estimated noise functions to the ground truth for TENOR and our extended baselines as shown in Fig. 4.

**Figure 4:** Comparison of the ground truth unseen temporal noise function $\boldsymbol{Q}(t)$ and its estimate $\hat{\boldsymbol{Q}}(t)$ from each *Temporal* method on synth. We only show the noise rate for the negative class for visual clarity.

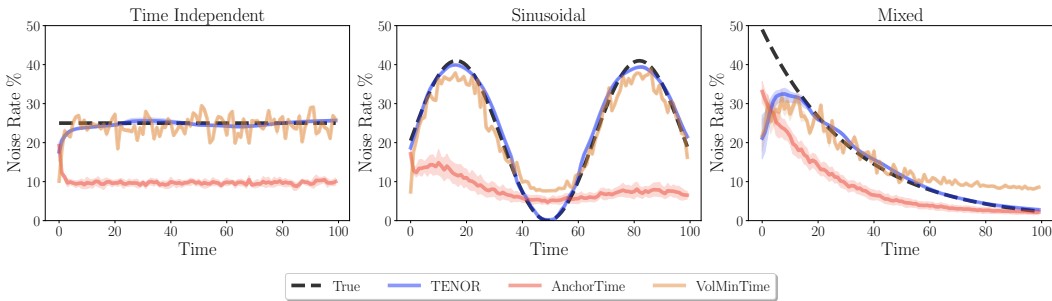

**Table 2:** Comparison of clean test Accuracy (%) of all methods with different temporal noise functions with 30% label noise on average. Dashed line separates *Static* and *Temporal* methods. $\pm$ is the st. dev. over 10 runs.

|  |  | Time Independent | | Exponential | | Linear | | Sigmoidal | | Sinusoidal | | Mixed | |
|---|---|---|---|---|---|---|---|---|---|---|---|---|---|
|  |  | Accuracy ↑ | MAE ↓ | Accuracy ↑ | MAE ↓ | Accuracy ↑ | MAE ↓ | Accuracy ↑ | MAE ↓ | Accuracy ↑ | MAE ↓ | Accuracy ↑ | MAE ↓ |
| Static | Uncorrected | 75.5±2.2 | – | 70.0±3.2 | – | 72.9±3.3 | – | 74.0±2.3 | – | 76.0±5.1 | – | 70.6±3.0 | – |
|  | Anchor | 86.2±2.7 | .04±.01 | 80.6±2.3 | .08±.01 | 78.1±2.4 | .12±.02 | 82.4±5.9 | .16±.01 | 82.0±3.6 | .15±.01 | 79.1±2.6 | .13±.01 |
|  | VolMinNet | 88.3±2.1 | .02±.01 | 81.3±2.0 | .07±.01 | 81.9±3.4 | .10±.02 | 81.6±2.5 | .14±.00 | 86.5±6.0 | .13±.01 | 85.1±2.7 | .11±.00 |
| Temporal | AnchorTime | 86.1±4.5 | .06±.01 | 80.6±2.5 | .08±.01 | 78.0±2.2 | .11±.01 | 80.7±2.7 | .14±.01 | 81.5±4.3 | .14±.01 | 80.0±1.8 | .11±.01 |
|  | VolMinTime | 86.7±1.6 | .08±.00 | 79.5±3.3 | .11±.01 | 81.8±4.2 | .10±.02 | 83.1±3.0 | .12±.01 | 85.3±5.7 | .11±.01 | 82.4±2.7 | .10±.00 |
|  | TENOR | **97.9±0.7** | **.01±.00** | **96.6±1.1** | **.03±.01** | **94.9±4.4** | **.03±.02** | **98.2±0.5** | **.02±.01** | **98.3±0.6** | **.03±.01** | **95.8±2.2** | **.03±.01** |

**Comparing Forward and Backward Loss Correction for Temporal Noise** Our results highlight important differences between the *forward sequence loss* and *backward sequence loss* . Here we assume we have perfect oracle knowledge of the noise function defined by $\boldsymbol{Q}(t)$. We notice that the *forward sequence loss* technique significantly outperforms the *backward sequence loss* technique across all types and amounts of label noise, despite both possessing similar statistical properties Fig. 2. When the label noise function is truly time-independent, both methods perform equally well. In settings with temporal label noise, we find consistent performance improvements using the *forward sequence loss* technique. In contrast, we find inconsistent effects for the *backward sequence loss* technique. For example, temporal modeling underperforms in the sinusoidal noise setting, but overperforms in the mixed noise. These results may stem from gradient-related issues with the *backward sequence loss* technique. Since the *backward sequence loss* technique requires an explicit inversion of the noise matrix at each time, the inverse-determinant of the matrix will scale the loss and therefore the gradients. We provide a more detailed analysis in Appendix D.

## 6 CONCLUSIONS

Noisy labels cause real problems for classification algorithms. Well-established methods exist to construct noise-tolerant classifiers in the static setting. However, in the time series domain, where labels are often observed sequentially, labels may be corrupted in a time-dependent fashion. For example, label quality may improve or worsen over time). To the best of our knowledge, building noise-tolerant classifiers in this setting has been unexplored. In order to remedy this, we first introduce a novel problem setting: *temporal label noise*. We show how we can learn provably robust classifiers from sequential data that have been corrupted with *temporal label noise* using knowledge of the underlying noise function. Existing methods on static data are ill-equipped to handle temporal label noise and as a result substantially under perform. In practice, however, the temporal noise function is unknown, so we propose a novel learning objective, TENOR, which can learn robust sequential classifiers with noisy labels without any prior assumptions about the unobserved temporal noise function. Through an extensive empirical study on synthetic and real-world data, our method leads to state-of-the-art performance in the presence of diverse, unseen temporal label noise functions.

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

Appendix

## A PROOFS

We make the following assumptions regarding the conditional time series distribution for the clean labels $p(\mathbf{y}_{1:T} \mid \mathbf{x}_{1:T})$ and noisy labels $p(\tilde{\mathbf{y}}_{1:T} \mid \mathbf{x}_{1:T})$:

We make the following assumptions about the clean data distribution:

**Assumption 2.** *The clean labels $\mathbf{y}_t$ at times $t = 1, \ldots, T$ are conditionally independent given the features observed up to time $t$ $\mathbf{x}_{1:t}$.*

$$\mathbf{y}_t \perp\!\!\!\perp \mathbf{y}_{1:t-1} \mid \mathbf{x}_{1:t} \tag{7}$$

**Assumption 3.** *The clean labels $\mathbf{y}_t$ at time $t$ is conditionally independent from $\mathbf{x}_{t+1}$ given $\mathbf{x}_{1:t}$:*

$$\boldsymbol{p}(\mathbf{y}_{1:T} \mid \mathbf{x}_{1:T}) = \prod_{t=1}^{\top} \boldsymbol{p}(\mathbf{y}_t \mid \mathbf{x}_{1:t}). \tag{8}$$

**Assumption 4.** *The noisy labels at time $t$ $\tilde{\mathbf{y}}_t$ are conditionally independent of $\mathbf{x}_{1:t}$ given the clean labels $\mathbf{y}_t$ at time $t$.*

$$\boldsymbol{p}(\tilde{\mathbf{y}}_t \mid \mathbf{x}_{1:t}) = \sum_{c=1}^{C} q_t(\tilde{\mathbf{y}}_t \mid \mathbf{y}_t = c) p(\mathbf{y}_t = c \mid \mathbf{x}_{1:t}) \tag{9}$$

Note that the following property follows from the above assumptions:

$$\boldsymbol{p}(\tilde{\mathbf{y}}_{1:T} \mid \mathbf{x}_{1:T}) = \prod_{t=1}^{\top} \boldsymbol{q}_t(\tilde{\mathbf{y}}_t \mid \mathbf{x}_{1:t}). \tag{10}$$

**Definitions** We start by defining some of the quantities that will be important for our forward and backward proofs:

$\boldsymbol{p}(\mathbf{y}_t \mid \mathbf{x}_{1:t}) := [p(\mathbf{y}_t = c \mid \mathbf{x}_{1:t})]_{c=1:C}^{\top}, \in \mathbb{R}^{C \times 1}$ (Vector of probabilities for each label value, for the clean label distribution)

$\boldsymbol{p}(\tilde{\mathbf{y}}_t \mid \mathbf{x}_{1:t}) := [p(\mathbf{y}_t = c \mid \mathbf{x}_{1:t})]_{c=1:C}^{\top} \in \mathbb{R}^{C \times 1}$ (Vector of probabilities for each possible label value, for the noisy label distribution)

$\boldsymbol{h}_\theta(\boldsymbol{x}_{1:t}) = \boldsymbol{p}_\theta(\mathbf{y}_t \mid \mathbf{x}_{1:t} = \boldsymbol{x}_{1:t}) : \mathbb{R}^{d \times t} \to \mathbb{R}^C$ (Classifier that predicts label distribution at $t$ given preceding observations)

$\boldsymbol{h}_\theta(\boldsymbol{x}_{1:t}) = \boldsymbol{\psi}^{-1}(\boldsymbol{g}_\theta(\boldsymbol{x}_{1:t}))$ (When $h_\theta$ is a deep network, $g_\theta$ is the final logits and $\psi : \Delta^{C-1} \to \mathbb{R}^C$ represents an invertible link function whose inverse maps the logits to a valid probability; i.e. a softmax function). We thus assume that

$\boldsymbol{Q}_t := [q_t(\tilde{\mathbf{y}}_t = k \mid \mathbf{y}_t = j)]_{j,k} \in \mathbb{R}^{C \times C}$ (The temporal noise matrix at time $t$)

$\ell_t(y_t, \boldsymbol{h}_\theta(\boldsymbol{x}_{1:t})) = -\log p_\theta(\mathbf{y}_t = y_t \mid \mathbf{x}_{1:t} = \boldsymbol{x}_{1:t}) : \mathcal{Y} \times \mathbb{R}^C \to \mathbb{R}$ (loss at $t$)

$\ell_{\psi,t}(y_t, \boldsymbol{h}_\theta(\boldsymbol{x}_{1:t})) = \ell_t(y_t, \boldsymbol{\psi}^{-1}\boldsymbol{h}_\theta(\boldsymbol{x}_{1:t}))$ (A composite loss function is a loss function that uses the aid of a link function: $\boldsymbol{\psi}$)

$\boldsymbol{\ell}_t(\boldsymbol{h}_\theta(\boldsymbol{x}_{1:t})) = [\ell_t(c, \boldsymbol{h}_\theta(\boldsymbol{x}_{1:t})]_{c=1:C}^{\top} : \mathbb{R}^C \to \mathbb{R}^C$ (vector of NLL losses, for each possible value of the ground truth)

$\overleftarrow{\boldsymbol{\ell}}_t(\boldsymbol{h}_\theta(\boldsymbol{x}_{1:t})) = \boldsymbol{Q}_t^{-1}\boldsymbol{\ell}(\boldsymbol{h}_\theta(\boldsymbol{x}_{1:t})) = [\overleftarrow{\ell}_t(c, \boldsymbol{h}_\theta(\boldsymbol{x}_{1:t}))]_{c=1:C}^\top : \mathbb{R}^C \to \mathbb{R}^C$ (vector of $\overleftarrow{\ell}_t(\cdot, \cdot)$ losses, for each possible value of the ground truth)

$\overleftarrow{\ell}_t(c, \boldsymbol{h}_\theta(\boldsymbol{x}_{1:t})) = [\boldsymbol{Q}_t^{-1}]_{c,:} \cdot \boldsymbol{\ell}(\boldsymbol{h}_\theta(\boldsymbol{x}_{1:t}))$

$\overrightarrow{\ell}_{t,\psi}(c, \boldsymbol{h}_\theta(\boldsymbol{x}_{1:t})) = \ell_t(c, \boldsymbol{Q}_t^\top \cdot \boldsymbol{\psi}^{-1}(\boldsymbol{g}_\theta))$

$\overrightarrow{\ell}_{seq,\psi}(\boldsymbol{y}_{1:T}, \boldsymbol{h}_\theta(\boldsymbol{x}_{1:t})) = \sum_{t=1}^T \overrightarrow{\ell}_{t,\psi}(c, \boldsymbol{h}_\theta(\boldsymbol{x}_{1:t}))$

**Lemma 1.** $\mathbb{E}_{\mathbf{y}_{1:T}|\mathbf{x}_{1:T}} \log p(\mathbf{y}_t \mid \mathbf{x}_{1:T}) = \mathbb{E}_{\mathbf{y}_t|\mathbf{x}_{1:t}} \log p(\mathbf{y}_t \mid \mathbf{x}_{1:t})$

*Proof.*

$$\mathbb{E}_{\mathbf{y}_{1:T}|\mathbf{x}_{1:T}} \log(p(\mathbf{y}_t \mid \mathbf{x}_{t:T})) = \sum_{c_1}\sum_{c_2}\cdots\sum_{c_T} p(\mathbf{y}_1 = c_1, \mathbf{y}_2 = c_2, \ldots \mathbf{y}_T = c_T \mid \mathbf{x}_{1:T})$$
$$log(p(\mathbf{y}_t = c_t \mid \mathbf{x}_{t:T}))$$

$$= \sum_{c}\sum_{c_1}\cdots\sum_{c_T} p(\mathbf{y}_1 = c_1, \mathbf{y}_2 = c_2, \ldots, \mathbf{y}_{t-1} = c_{t-1} \mid \mathbf{x}_{1:T})$$
$$* p(\mathbf{y}_{t+1} = c_{t+1}, \ldots, \mathbf{y}_T = c_T \mid \mathbf{x}_{1:T}) * p(\mathbf{y}_t = c_t \mid \mathbf{x}_{1:T})$$
$$* \log(p(\mathbf{y}_t = c_t \mid \mathbf{x}_{t:T}))$$

$$= \sum_{c_1}\sum_{c_2}\cdots\sum_{c_T} p(\mathbf{y}_1 = c_1, \mathbf{y}_2 = c_2, \ldots, \mathbf{y}_{t-1} = c_{t-1} \mid \mathbf{x}_{1:t-1})$$
$$* p(\mathbf{y}_{t+1} = c_{t+1}, \ldots, \mathbf{y}_T = c_T \mid \mathbf{x}_{t+1:T}) * p(\mathbf{y}_t = c_t \mid \mathbf{x}_{1:t})$$
$$* \log(p(\mathbf{y}_t = c_t \mid \mathbf{x}_{1:t}))$$

$$= \mathbb{E}_{\mathbf{y}_{1:t-1}|\mathbf{x}_{1:t-1}} \mathbb{E}_{\mathbf{y}_{t+1:T}|\mathbf{x}_{t+1:T}} p(\mathbf{y}_t = c_t \mid \mathbf{x}_{1:t}) \log(p(\mathbf{y}_t = c_t \mid \mathbf{x}_{1:t}))$$

$$= \mathbb{E}_{\mathbf{y}_{1:t-1}|\mathbf{x}_{1:t-1}} \mathbb{E}_{\mathbf{y}_{t+1:T}|\mathbf{x}_{t+1:T}} \left[\mathbb{E}_{\mathbf{y}_t|\mathbf{x}_{1:t}} \log(p(\mathbf{y}_t \mid \mathbf{x}_{1:t}))\right]$$
(Note that $\left[\mathbb{E}_{p(\mathbf{y}_t|\mathbf{x}_{1:T})} \log(p(\mathbf{y}_t \mid \mathbf{x}_{1:t}))\right]$ is constant )

$$= \mathbb{E}_{p(\mathbf{y}_t|\mathbf{x}_{1:t})} \log(p(\mathbf{y}_t \mid \mathbf{x}_{1:t}))$$
(From property that $\mathbb{E}[C] = C$ for constant $C$

$\square$

**Lemma 2.** $\operatorname{argmax}_\theta \mathbb{E}_{\mathbf{y}_{1:T}|\mathbf{x}_{1:T}} \log\left(\boldsymbol{p}_\theta(\mathbf{y}_{1:T} \mid \mathbf{x}_{1:T})\right) = \operatorname{argmin}_\theta \sum_{t=1}^T \mathbb{E}_{\tilde{\mathbf{y}}_t|\mathbf{x}_{1:t}} \overleftarrow{\ell}_t(\tilde{\mathbf{y}}_t, \boldsymbol{h}_\theta(\mathbf{x}_{1:t}))$

*Proof.*

$$\operatorname*{argmax}_{\theta} \mathbb{E}_{\mathbf{y}_{1:T}|\mathbf{x}_{1:T}} \log \left( \boldsymbol{p}_\theta(\mathbf{y}_{1:T} \mid \mathbf{x}_{1:T}) \right)$$

$$= \operatorname*{argmax}_{\theta} \mathbb{E}_{\mathbf{y}_{1:T}|\mathbf{x}_{1:T}} \log \big( \prod_{t=1}^{T} \boldsymbol{p}_\theta(\mathbf{y}_t \mid \mathbf{x}_{1:t}) \big) \qquad \text{(by Assumption 2)}$$

$$= \operatorname*{argmax}_{\theta} \mathbb{E}_{\mathbf{y}_{1:T}|\mathbf{x}_{1:T}} \sum_{t=1}^{T} \log \left( \boldsymbol{p}_\theta(\mathbf{y}_t \mid \mathbf{x}_{1:t}) \right)$$

$$= \operatorname*{argmax}_{\theta} \sum_{t=1}^{T} \mathbb{E}_{\mathbf{y}_{1:T}|\mathbf{x}_{1:T}} \log \left( \boldsymbol{p}_\theta(\mathbf{y}_t \mid \mathbf{x}_{1:t}) \right) \qquad \text{(due to linearity of } \mathbb{E})$$

$$= \operatorname*{argmax}_{\theta} \sum_{t=1}^{T} \mathbb{E}_{\mathbf{y}_t|\mathbf{x}_{1:t}} \log \left( \boldsymbol{p}_\theta(\mathbf{y}_t \mid \mathbf{x}_{1:t}) \right) \qquad \text{(by Lemma 1)}$$

$$= \operatorname*{argmin}_{\theta} \sum_{t=1}^{T} \mathbb{E}_{\mathbf{y}_t|\mathbf{x}_{1:t}} \ell_t(\mathbf{y}_t, \boldsymbol{h}_\theta(\mathbf{x}_{1:t})) \qquad \text{(by definition of } \ell_t)$$

$$= \operatorname*{argmin}_{\theta} \sum_{t=1}^{T} \boldsymbol{p}(\mathbf{y}_t \mid \mathbf{x}_{1:t})^\top \boldsymbol{\ell}_t(\boldsymbol{h}_\theta(\mathbf{x}_{1:t}))$$

$$= \operatorname*{argmin}_{\theta} \sum_{t=1}^{T} \boldsymbol{p}(\tilde{\mathbf{y}}_t \mid \mathbf{x}_{1:t})^\top \boldsymbol{Q}_t^{-1} \boldsymbol{\ell}_t(\boldsymbol{h}_\theta(\mathbf{x}_{1:t}))$$

$$= \operatorname*{argmin}_{\theta} \sum_{t=1}^{T} \boldsymbol{p}(\tilde{\mathbf{y}}_t \mid \mathbf{x}_{1:t})^\top \overleftarrow{\boldsymbol{\ell}}_t(\boldsymbol{h}_\theta(\mathbf{x}_{1:t}))$$

$$= \operatorname*{argmin}_{\theta} \sum_{t=1}^{T} \mathbb{E}_{\tilde{\mathbf{y}}_t|\mathbf{x}_{1:t}} [\overleftarrow{\ell}_t(\tilde{\mathbf{y}}_t, \boldsymbol{h}_\theta(\mathbf{x}_{1:t}))]$$

$\square$

**Theorem 3.** *Let* $\overleftarrow{\ell}_{seq}(\tilde{\boldsymbol{y}}_{1:T}, \boldsymbol{h}_\theta(\boldsymbol{x}_{1:T})) = \sum_{t=1}^{T} \overleftarrow{\ell}_t(\tilde{y}_t, \boldsymbol{h}_\theta(\boldsymbol{x}_{1:t}))$. *Then,*

$$\operatorname*{argmax}_{\theta} \mathbb{E}_{\mathbf{y}_{1:T}|\mathbf{x}_{1:T}} \log \left( p_\theta(\mathbf{y}_{1:T} \mid \mathbf{x}_{1:T}) \right) = \operatorname*{argmin}_{\theta} \mathbb{E}_{\tilde{\mathbf{y}}_{1:T}|\mathbf{x}_{1:T}} \overleftarrow{\ell}_{seq}(\tilde{\mathbf{y}}_{1:T}, \boldsymbol{h}_\theta(\mathbf{x}_{1:T}))$$

*Proof.*

$$\operatorname*{argmin}_{\theta} \mathbb{E}_{\tilde{\mathbf{y}}_{1:T}|\mathbf{x}_{1:T}} \overleftarrow{\ell}_t(\tilde{\mathbf{y}}_{1:T}, \mathbf{h}_\theta(\mathbf{x}_{1:T})) = \operatorname*{argmin}_{\theta} \mathbb{E}_{\tilde{\mathbf{y}}_{1:T}|\mathbf{x}_{1:T}} \sum_{t=1}^{T} \overleftarrow{\ell}_t(\tilde{\mathbf{y}}_t, \boldsymbol{h}_\theta(\mathbf{x}_{1:t}))$$

$$= \operatorname*{argmin}_{\theta} \sum_{t=1}^{T} \mathbb{E}_{\tilde{\mathbf{y}}_{1:T}|\mathbf{x}_{1:T}} \overleftarrow{\ell}_t(\tilde{\mathbf{y}}_t, \boldsymbol{h}_\theta(\mathbf{x}_{1:t}))$$

$$= \operatorname*{argmin}_{\theta} \sum_{t=1}^{T} \mathbb{E}_{\tilde{\mathbf{y}}_t|\mathbf{x}_{1:t}} \overleftarrow{\ell}_t(\tilde{\mathbf{y}}_t, \boldsymbol{h}_\theta(\mathbf{x}_{1:t})) \qquad \text{(by Lemma 1)}$$

$$= \operatorname*{argmax}_{\theta} \mathbb{E}_{\mathbf{y}_{1:T}|\mathbf{x}_{1:T}} \log \left( \boldsymbol{p}_\theta(\mathbf{y}_{1:T} \mid \mathbf{x}_{1:T}) \right)$$

$$\text{(by Lemma 2)}$$

$\square$

**Theorem 4.** $\operatorname*{argmin}_{\theta} \mathbb{E}_{\tilde{\mathbf{y}}_{1:T}, \mathbf{x}_{1:T}} \overrightarrow{\ell}_{seq,\psi}(\mathbf{y}_{1:T}, \boldsymbol{g}_\theta(\mathbf{x}_{1:T})) = \operatorname*{argmin}_{\theta} \sum_{t=1}^{T} \mathbb{E}_{\mathbf{y}_{1:t}, \mathbf{x}_{1:t}} \ell_{t,\phi}(\mathbf{y}_{1:T}, \boldsymbol{g}_\theta(\mathbf{x}_{1:T})).$

*Proof.* First, note that:

$$\overrightarrow{\ell}_{t,\psi}(\mathbf{y}_t, \boldsymbol{h}_\theta(\mathbf{x}_{1:t})) = \ell_t(\mathbf{y}_t, \boldsymbol{Q}_t^\top \boldsymbol{\psi}^{-1}(\boldsymbol{g}_\theta(\mathbf{x}_{1:t}))) \tag{11}$$

$$= \ell_{\phi_t,t}(\mathbf{y}_t, \boldsymbol{g}_\theta(\mathbf{x}_{1:t})), \tag{12}$$

where $\boldsymbol{\phi}_t^{-1} = \boldsymbol{\psi}^{-1} \circ \boldsymbol{Q}_t^\top$. Thus, $\boldsymbol{\phi}_t : \Delta^{C-1} \to \mathbb{R}^C$ is invertible, and is thus a proper composite loss [40].

Thus, as shown in Patrini et al. [35]:

$$\operatorname*{argmin}_\theta \mathbb{E}_{\tilde{\mathbf{y}}_t, \mathbf{x}_{1:t}} \ell_{\phi,t}(\mathbf{y}_t, \boldsymbol{g}_\theta(\mathbf{x}_{1:t})) = \operatorname*{argmin}_\theta \mathbb{E}_{\tilde{\mathbf{y}}_t|\mathbf{x}_{1:t}} \ell_{\phi,t}(\mathbf{y}_t, \boldsymbol{g}_\theta(\mathbf{x}_{1:t})) \tag{13}$$

$$= \boldsymbol{\phi}_t(\boldsymbol{p}(\tilde{\mathbf{y}}_t \mid \mathbf{x}_{1:t})) \qquad \text{(property of proper composite losses)}$$

$$= \boldsymbol{\psi}((\boldsymbol{Q}_t^{-1})^\top \boldsymbol{p}(\tilde{\mathbf{y}}_t \mid \mathbf{x}_{1:t}))) \tag{14}$$

$$= \boldsymbol{\psi}(\boldsymbol{p}(\mathbf{y}_t \mid \mathbf{x}_{1:t})) \tag{15}$$

The above holds for the minimizer at a single time step, not the sequence as a whole. To find the minimizer of the loss over the entire sequence:

$$\operatorname*{argmin}_\theta \mathbb{E}_{\mathbf{x}_{1:T},\tilde{\mathbf{y}}_{1:T}} \overrightarrow{\ell}_{seq,\psi}(\tilde{\mathbf{y}}_{1:T}, \boldsymbol{g}_\theta(\mathbf{x}_{1:T})) = \operatorname*{argmin}_\theta \mathbb{E}_{\tilde{\mathbf{y}}_{1:T}|\mathbf{x}_{1:T}} \overrightarrow{\ell}_{seq,\psi}(\tilde{\mathbf{y}}_{1:T}, \boldsymbol{g}_\theta(\mathbf{x}_{1:T})) \tag{16}$$

$$= \operatorname*{argmin}_\theta \mathbb{E}_{\tilde{\mathbf{y}}_{1:T}|\mathbf{x}_{1:T}} \sum_{t=1}^T \overrightarrow{\ell}_{t,\psi}(\tilde{\mathbf{y}}_t, \boldsymbol{g}_\theta(\mathbf{x}_{1:t})) \tag{17}$$

$$= \operatorname*{argmin}_\theta \sum_{t=1}^T \mathbb{E}_{\tilde{\mathbf{y}}_{1:T}|\mathbf{x}_{1:T}} \overrightarrow{\ell}_{t,\psi}(\tilde{\mathbf{y}}_t, \boldsymbol{g}_\theta(\mathbf{x}_{1:t})) \tag{18}$$

$$= \operatorname*{argmin}_\theta \sum_{t=1}^T \mathbb{E}_{\tilde{\mathbf{y}}_t|\mathbf{x}_{1:t}} \overrightarrow{\ell}_{t,\psi}(\tilde{\mathbf{y}}_t, \boldsymbol{g}_\theta(\mathbf{x}_{1:t})) \tag{19}$$

$$= \operatorname*{argmin}_\theta \sum_{t=1}^T \mathbb{E}_{\tilde{\mathbf{y}}_t|\mathbf{x}_{1:t}} \ell_{t,\phi}(\tilde{\mathbf{y}}_t, \boldsymbol{g}_\theta(\mathbf{x}_{1:t})) \tag{20}$$

As the minimizer of the sum will be the function that minimizes each element of the sum, then $\operatorname{argmin}_\theta \mathbb{E}_{\tilde{\mathbf{y}}_{1:T},\mathbf{x}_{1:T}} \overrightarrow{\ell}_{seq,\psi}(\mathbf{y}_{1:T}, \boldsymbol{g}_\theta(\mathbf{x}_{1:T})) = \boldsymbol{\psi}(\boldsymbol{p}(\mathbf{y}_{1:T} \mid \mathbf{x}_{1:T}))$. Note that the $\operatorname{argmin}_\theta \sum_{t=1}^T \mathbb{E}_{\mathbf{y}_{1:t},\mathbf{x}_{1:t}} \ell_{t,\phi}(\mathbf{y}_{1:T}, \boldsymbol{g}_\theta(\mathbf{x}_{1:T})) = \boldsymbol{\psi}(\boldsymbol{p}(\mathbf{y}_{1:T} \mid \mathbf{x}_{1:T}))$, because the minimizer of the NLL is the data distribution. Thus, $\operatorname{argmin}_\theta \mathbb{E}_{\tilde{\mathbf{y}}_{1:T},\mathbf{x}_{1:T}} \overrightarrow{\ell}_{seq,\psi}(\mathbf{y}_{1:T}, \boldsymbol{g}_\theta(\mathbf{x}_{1:T})) = \operatorname{argmin}_\theta \sum_{t=1}^T \mathbb{E}_{\mathbf{y}_{1:t},\mathbf{x}_{1:t}} \ell_{t,\phi}(\mathbf{y}_{1:T}, \boldsymbol{g}_\theta(\mathbf{x}_{1:T}))$.

□

# B TENOR LEARNING ALGORITHM

We summarize the augmented Lagrangian approach to solving the TENOR objective in Algorithm 1

For all experiments we set $\lambda = 1, c = 1, \gamma = 2$, and $\eta = 2$. $k$ and the maximum number of SGD iterations are set to 15 and 10, respectively. This is to ensure that the total number of epochs is 150, which is the max number of epochs used for all experiments.

# C LIMITATIONS AND FUTURE WORK

Our methods rely on several assumptions to ensure conditional independence between labels across time (Assumption 3). Though these are standard assumptions in sequential modelling, these assumptions could be relaxed in settings where the the joint distribution of a sequence does not factor out

---

**Algorithm 1** TENOR Learning Algorithm

---

**Input:** Noisy Training Dataset $D$, hyperparameters $\gamma$ and $\eta$
**Output:** Model $\theta$, Temporal Noise Function $\omega$

   $c \leftarrow 1$ and $\lambda \leftarrow 1$
   **for** $k = 1, 2, 3, ...,$ **do**
      $\theta^k, \omega^k = \arg\min_{\theta,\omega} \mathcal{L}(\theta, \omega)$             $\triangleright$ Computed with SGD using the Adam optimizer
      $\lambda \leftarrow \lambda + c * R_t(\theta^k, \omega^k)$               $\triangleright$ Update Lagrange multiplier
      **if** $k > 0$ and $R_t(\theta^k, \omega^k) > \gamma R_t(\theta^{k-1}, \omega^{k-1})$ **then**
         $c \leftarrow \eta c$
      **else**
         $c \leftarrow c$
      **end if**
      **if** $R_t(\theta^k, \omega^k) == 0$ **then**
         break
      **end if**
   **end for**

---

this way. Existing work in Empirical Risk Minimization (ERM) on highly dependent sequences may serve as a promising direction [31, 32, 43]. Leveraging these results may help us to also provide finite-sample guarantees. Though we provide equivalence statements in Theorem 1 and Theorem 2, we understand the importance of finite-sample guarantees as we are using ERM on a single-draw of the noisy distribution. Using the equivalences we derive, finite-sample guarantees and excess-risk bounds may be drawn from existing results.

Another direction to explore is to relax the assumption that all samples have the same unerlying label noise function. This may be a result of different labellers with different labelling proficiency over time.

Lastly, we intentionally leave our Definition 1 generalizable to continuous time models. In this work, we represented $Q_\omega$ as a fully connected feed-forward neural network. In general, however, $Q_\omega$ can be parameterized however a user sees fit – e.g., as a time-dependent noise function could be with a Gaussian Process or a neural ODE that can learn $\frac{dQ}{dt}$ and generate $\boldsymbol{Q}_t$ across continuous time $1, \ldots, T$ by solving an initial value problem.

# D  EXPERIMENTAL DETAILS

| Dataset | Classification Task | $n$ | $d$ | $T$ |
|---|---|---|---|---|
| eeg_eye [42] | Eye Open vs Eye Closed | 299 | 14 | 50 |
| eeg_sleep [17] | Sleep vs Awake | 964 | 7 | 100 |
| har [41] | Walking vs Not Walking | 192 | 9 | 50 |
| har70 [29] | Walking vs Not Walking | 444 | 6 | 100 |
| synth | [describe model in notation] | 1,000 | 50 | 100 |

**Table 3:** Datasets used in the experiments. Classification tasks, number of samples ($n$), dimensionality at each time step ($d$), and sequence length ($T$) are shown.

## D.1  DATASET DETAILS

**Synthetic** We generate data for binary and multiclass classification with $n = 1\,000$ samples and $d = 50$ features over $T = 100$ time steps. We generate the class labels and obvservations for each time step using a Hidden Markov Model (HMM). The transition matrix generating the markov chain is uniform ensuring an equal likelihood of any state at any given time. We corrupted them using multidimensional (50) Gaussian emissions. The mean of the gaussian for state/class $c$ is set to $c$ with variance $1.5$ (i.e. class 1 has mean 1 and variance 1.5). The high-dimensionality and overlap in feature-space between classes makes this a sufficiently difficult task, especially under label noise. We use a batchsize of 256

**HAR** from UC Irvine [41] consists of inertial sensor readings of 30 adult subjects performing activities of daily living. The sensor signals are already preprocessed and a vector of features at each time step are provided. We apply z-score normalization at the participant-level, then split the dataset into subsequences of a fixed size 50. We use a batchsize of 64.

**HAR70** from UC Irvine [29] consists of inertial sensor readings of 18 elderly subjects performing activities of daily living. The sensor signals are already preprocessed and a vector of features at each time step are provided. We apply z-score normalization at the participant-level, then split the dataset into subsequences of a fixed size 100. We use a batchsize of 256.

**EEG SLEEP** from Physionet [17] consists of EEG data measured from 197 different whole nights of sleep observation, including awake periods at the start, end, and intermittently. We apply z-score normalization at the whole night-level. Then downsample the data to have features and labels each minute, as EEG data is sampled at 100Hz and labels are sampled at 1Hz. We then split the data into subsequences of a fixed size 100. We use a batchsize of 512.

**EEG EYE** from UC Irvine [42] consists of data measured from one continuous participant tasked with opening and closing their eyes while wearing a headset to measure their EEG data . We apply z-score normalization for the entire sequence, remove outliers (>5 SD away from mean), and split into subsequences of a fixed size 50. We use a batchsize of 128.

### D.2 SPECIFIC IMPLEMENTATION DETAILS

**GRU** the GRU $r : \mathbb{R}^d \times \mathbb{Z} \rightarrow \mathbb{R}^C \times \mathbb{Z}$ produces an *output vector* such that the output of $r(\boldsymbol{x}_t, \boldsymbol{z}_{t-1})$ is our model for $\boldsymbol{h}_\theta(\boldsymbol{x}_{1:t})$, and a *hidden state* $\boldsymbol{z}_t \in \mathbb{Z}$ that summarizes $\boldsymbol{x}_{1:t}$. We use a softmax activation on the output vector of the GRU to make it a valid parameterization of $\boldsymbol{p}_\theta(y_t \mid \boldsymbol{x}_{1:t})$. The GRU has a single hidden layer with a 32 dimension hidden state.

**TENOR** TENOR uses an additional fully-connected neural network with 10 hidden layers that outputs a $C * C$-dimensional vector to represent each entry of a flattened $\hat{\boldsymbol{Q}}_t$. To ensure the output of this network is valid for Definition 1, we reshape the prediction to be $C \times C$, apply a row-wise softmax function, add this to the identity matrix to ensure diagonal dominance, then rescale the rows to be row-stochastic. These operations are all differentiable, ensuring we can optimize this network with standard backpropagation.

**VolMinNet and VolMinTime** We do a similar parameterization for VolMinNet and VolMinTime, using a set of differentiable weights to represent the entries of $\boldsymbol{Q}_t$ rather than a neural network.

**Anchor and AnchorTime** Patrini et al. [35] show that in practice taking the 97th percentile anchor points rather than the maximum yield better results, so we use that same approach in our experiments. They also describe a two-stage approach: 1) estimate the anchor points after a warmup period 2) use the anchor points to train the classifier with forward corrected loss. We set the warmup period to 25 epochs.

### D.3 EXPERIMENTAL PARAMETERS

Given that the learning algorithm only has access to a noisy training dataset and performance is evaluated on a clean test set, a validation set must be drawn from clean test data or by manually cleaning the noisy training dataset which may be impractical. This makes hyperparameter tuning difficult in noisy label learning. As the optimal set of hyperparameters within each could vary for each method, noise type, amount of noise, and dataset, this represents a difficult task. To be fair for our experimental evaluations, we use the same set of hyperparameters for experiment, and only manually set batch size for each dataset.

Each model was trained for 150 epochs using the adam optimizer with default parameters and a learning rate of 0.01.

For VolMinNet, VolMinTime, and TENOR we use adam optimizer with default parameters and a learning rate of 0.01 to optimize each respective $\hat{\boldsymbol{Q}}_t$-estimation technique. $\lambda$ was set to $1e-4$ for VolMinNet and VolMinTime for all experiments, based on what was published previously [25] .

### D.4 NOISE INJECTION

To the best of our knowledge there are no noisy label time series datasets (i.e.: standardized datasets with both clean and noisy labels) to evaluate our methods. In line with prior experimental approaches, we propose a noise injection strategy which assumes some temporal noise function that can give us a noisy distribution to evaluate from. We deliberately pick a wide variety of noise types, varying the amount and functional form of time-dependent noise, including static noise setting (uniform noise at every time, akin to what baseline methods assume), and class-dependent noise structure Fig. 5.

**Figure 5:** Temporal functions that can be specified using a temporal label noise function $\boldsymbol{Q}(t)$. We present six examples for binary classification task (from top-left clockwise): time independent, exponential decay sinusoidal noise, mixed class-dependent noise, linear decay noise, sigmoid increasing noise. Each plot shows the off-diagonal entries of various parameterized forms of $\boldsymbol{Q}(t)$.

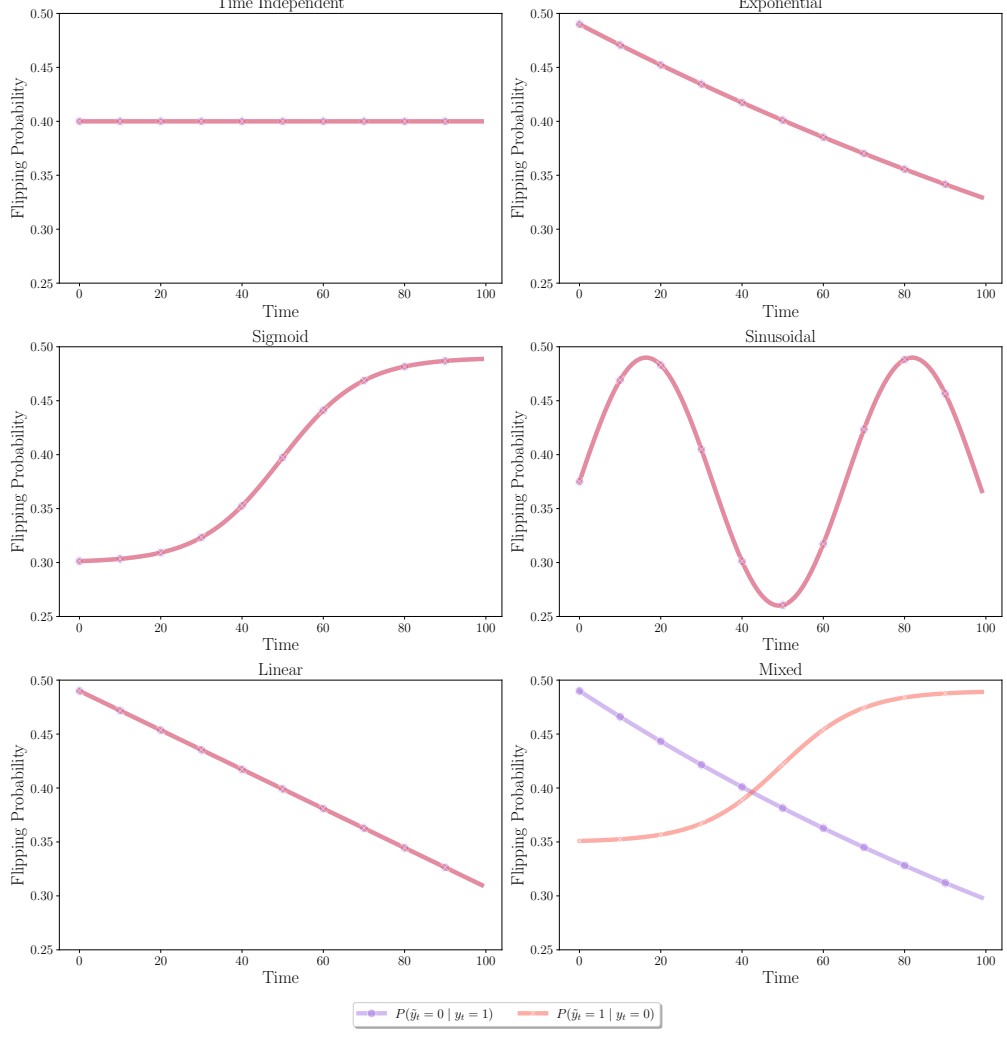

# E    COMPLETE RESULTS

## E.1    FORWARD VS BACKWARD LOSS

Here we explain some counter-intuitive results observed when comparing the *backward sequence loss* and *forward sequence loss* in static and temporal noise. We can understand this behaviour by realizing that *backward sequence loss* requires an explicit matrix-inversion which multiplies into your uncorrected loss term. This involves an inverse-determinant term that scales the loss at every time step. Consider the $C = 2$ setting, the magnitude of this term is controlled by the product of the off-diagonal entries - i.e: it approaches $\infty$ as the noise for each class in $\boldsymbol{Q}_t$ approaches the upper bound of noise $0.5$. This is why in Fig. 2, *backward sequence loss* performs particularly poorly in high noise. In the static case, where we average $\boldsymbol{Q}_t$ over time, we bound the inverse-determinant at each time step, therefore controlling these gradients by reducing the effect of any high noise time steps. In the Mixed noise setting in Fig. 2, we increase the noise in one class while decreasing the noise in the other over time, so the off-diagonal entries in $\boldsymbol{Q}_t$ never reach their upper bound at the same time. As a result this again provides a mechanism to control the inverse-determinant.

We provide complete comparisons of *backward sequence loss* and *forward sequence loss* across varying degrees of noise and all temporal noise functions in Fig. 6.

**Figure 6:** Comparing performance of models trained with *backward sequence loss* and *forward sequence loss* on `synth` with varying degrees of temporal label noise using either the true temporal noise function (Temporal) or the average temporal noise function (Static). Error bars are st. dev. over 10 runs.

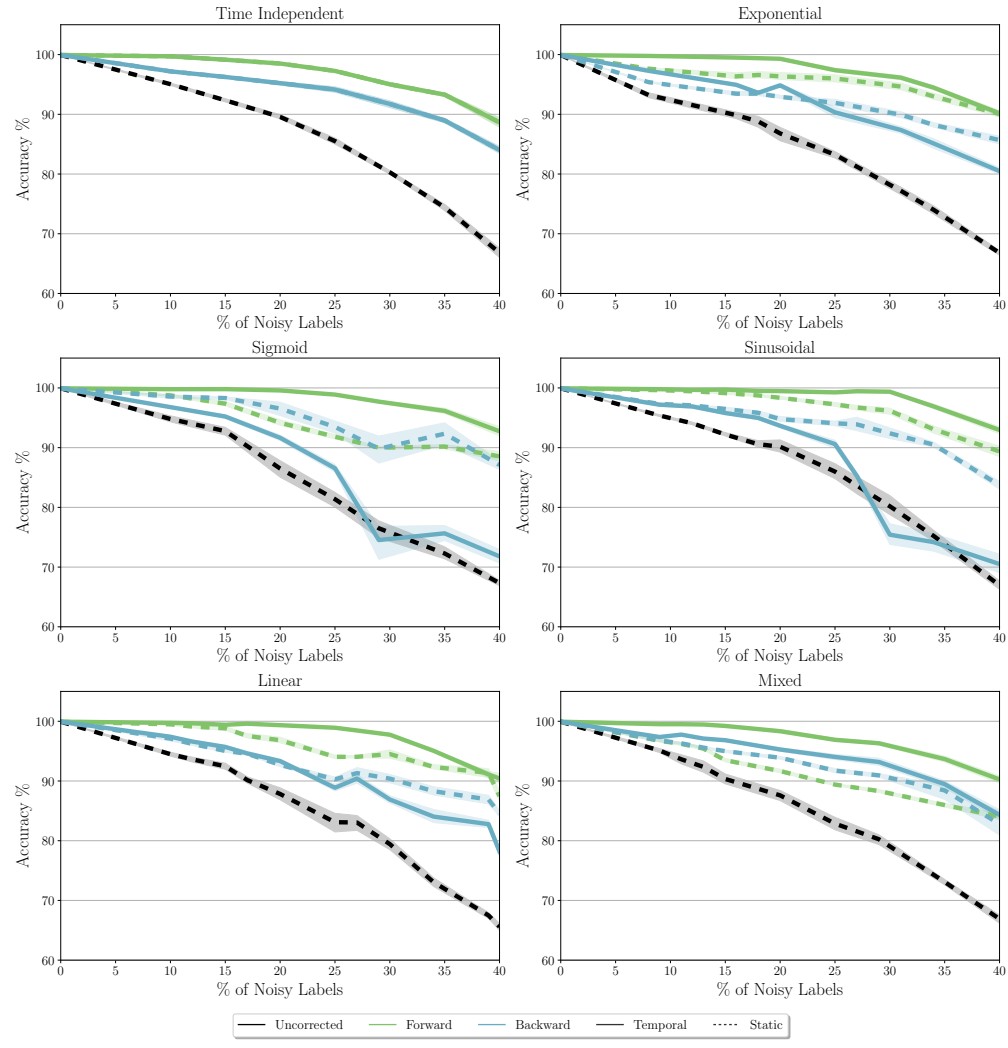

### E.2 REAL TEMPORAL LABEL NOISE

Prior work in the static noisy label literature typically aim to demonstrate the effectiveness of their methods on a real world noisy dataset, where the noise function is not imposed by the researcher. The primary dataset used is the `Clothing1M` dataset [53]. Despite containing real label noise, Clothing1M is inapplicable in our setting: it is not sequential data and each instance has only one label. In the spirit of evaluating TENOR on real-world noisy labels, we discovered and experimented with `extrasensory`, a noisy-labelled time series dataset [47]. `extrasensory` includes human activity data from smartphones and smartwatches collected from 60 users spread across 300,000 minutes of measurements. In contrast to `har` and `har70` (datasets originally used in our paper), `extrasensory` has no expert-labelled annotations, all the labels are user-provided and therefore are highly noisy. Users often misreport falling asleep and waking up, so we expect particularly high label noise during sleep/awake transitions

In order to identify the label noise in this sequential data, we partition and center the dataset from all users around sleep/awake transition periods. That is, for a fixed length window of 50, sleep/awake transitions occur around the $t = 25$ point. We then train our TENOR objective with the same model architecture and hyperparameters as above to classify sleep and awake over time. Since there are no 'clean' labels, we demonstrate that TENOR successfully identifies an interpretable temporal noise function.

In Fig. 7, we see TENOR predicts there exists higher label noise near sleep/awake transitions (around $t = 25$). We hope our work also encourages the community to seek further sources of real temporal noise.

**Figure 7:** TENOR-estimated $\hat{Q}_t$ for `extrasensory`. Error bars are st. dev. over 10 runs.

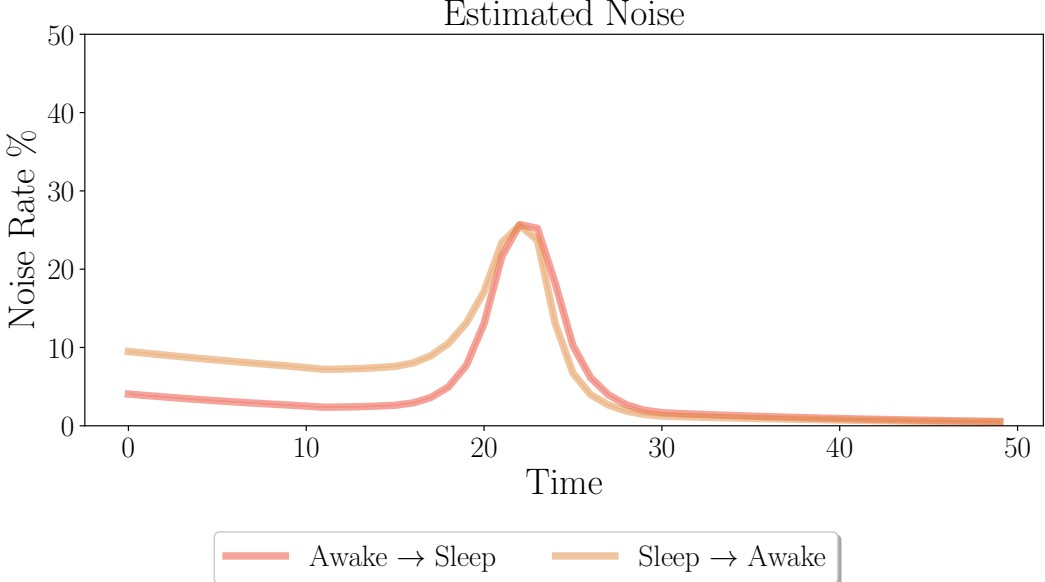

### E.3 ACCURACY OVER VARYING DEGREES NOISE

### E.4 MAE OVER VARYING DEGREES NOISE

**Figure 8:** Comparison of clean test set Accuracy (%) for `synth` across varying degrees of temporal label noise comparing all methods. Error bars are st. dev. over 10 runs.

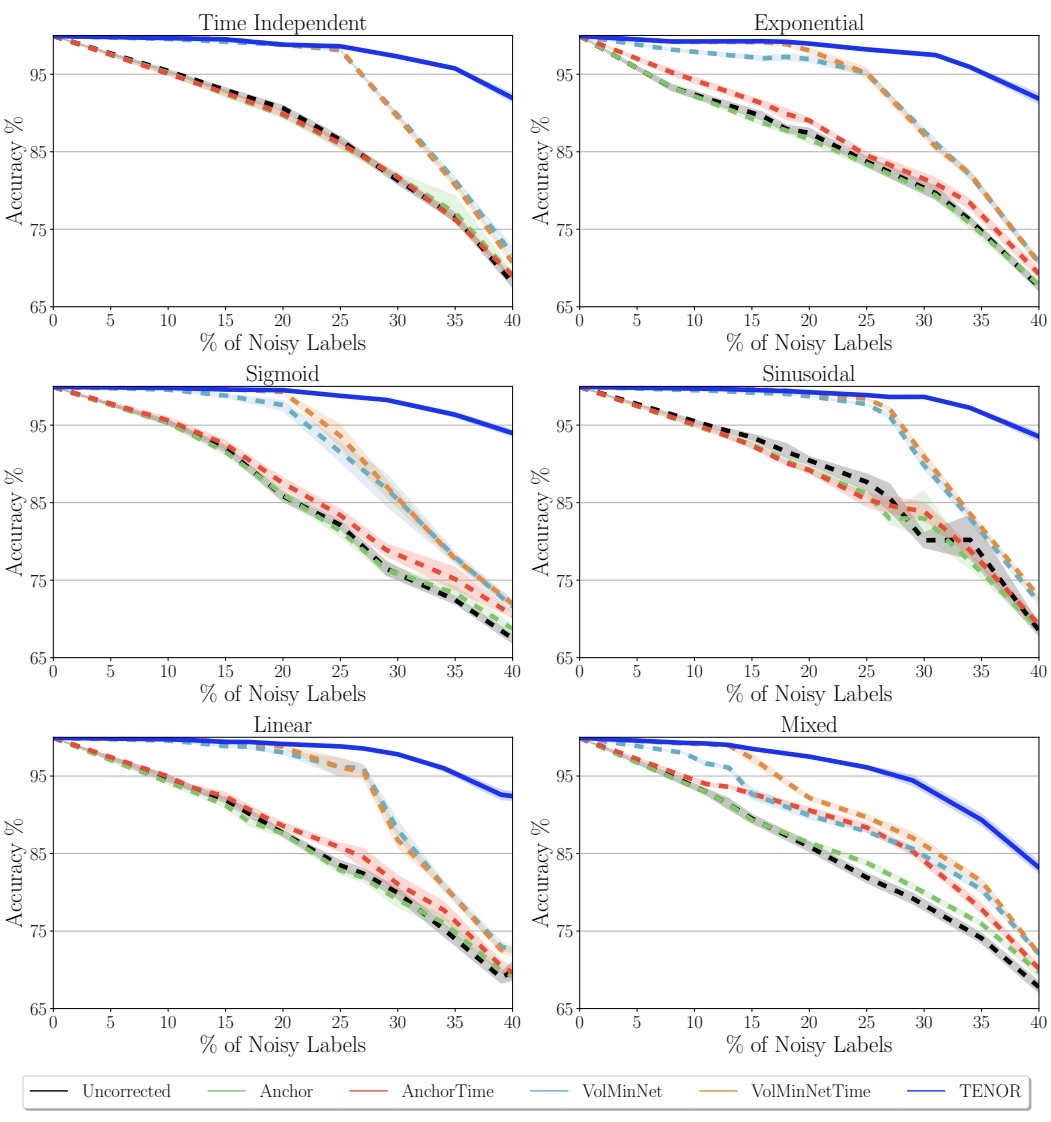

**Figure 9:** Comparison of clean test set Accuracy (%) for `har` across varying degrees of temporal label noise comparing all methods. Error bars are st. dev. over 10 runs.

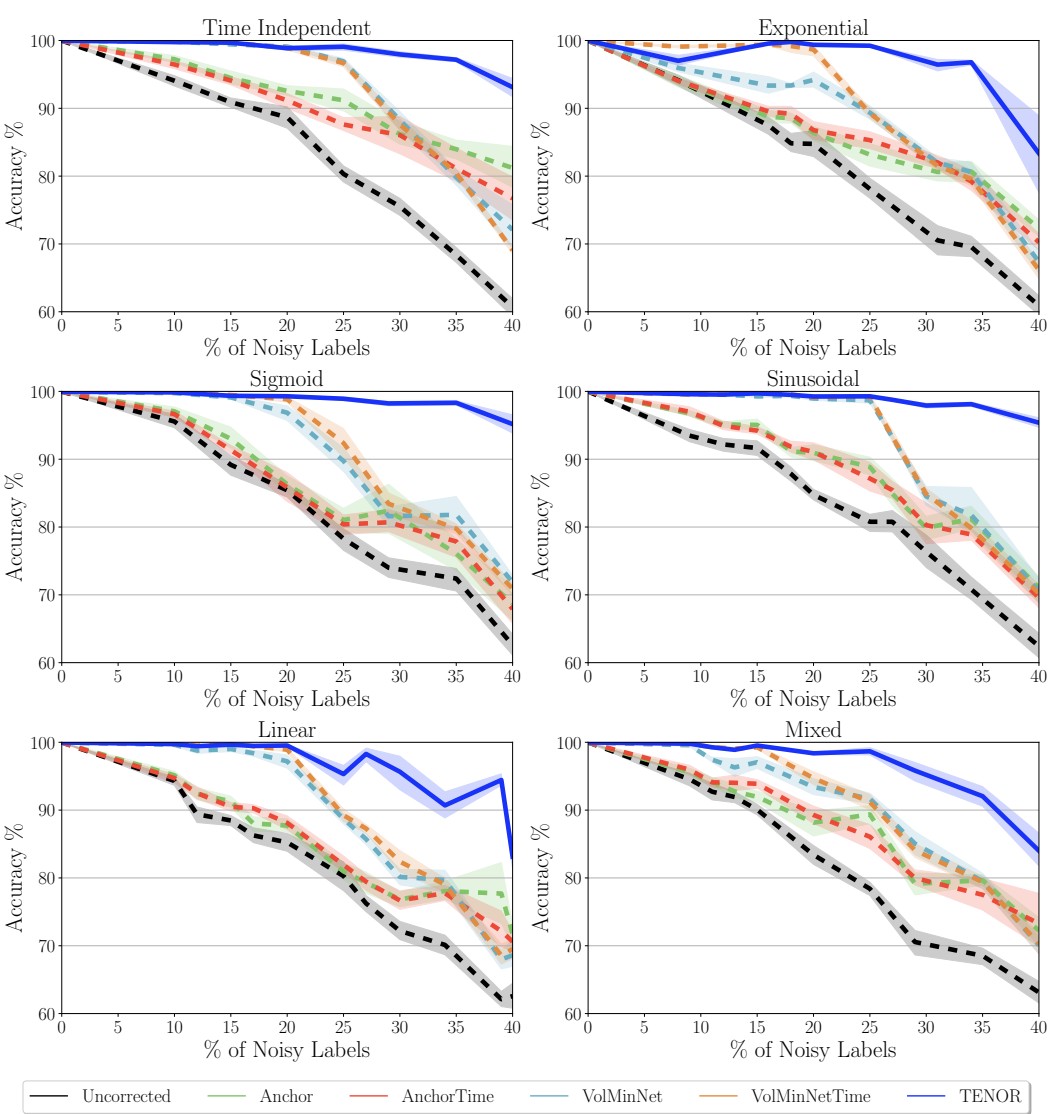

**Figure 10:** Comparison of clean test set Accuracy (%) for `har70` across varying degrees of temporal label noise comparing all methods. Error bars are st. dev. over 10 runs.

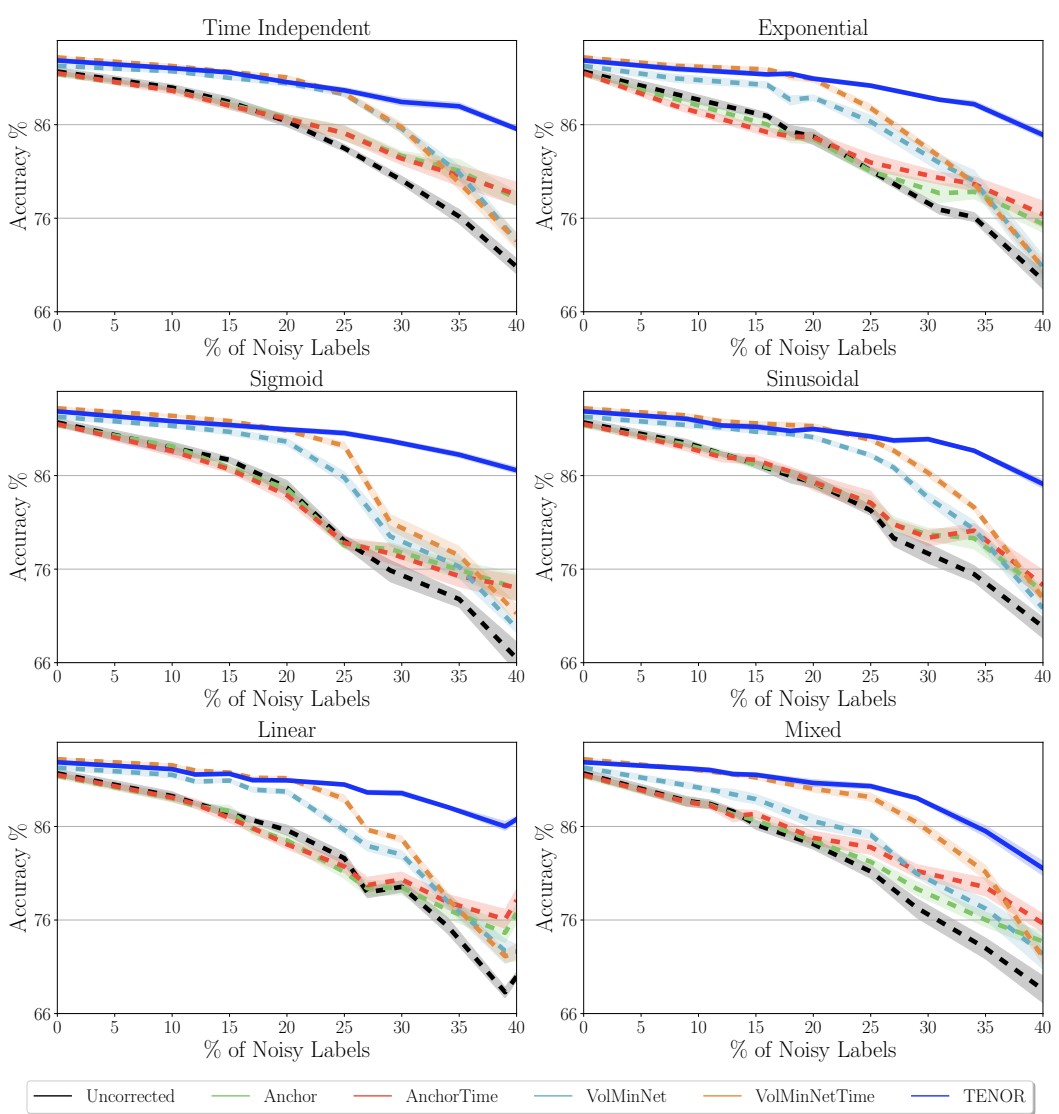

**Figure 11:** Comparison of clean test set Accuracy (%) for `eeg_sleep` across varying degrees of temporal label noise comparing all methods. Error bars are st. dev. over 10 runs.

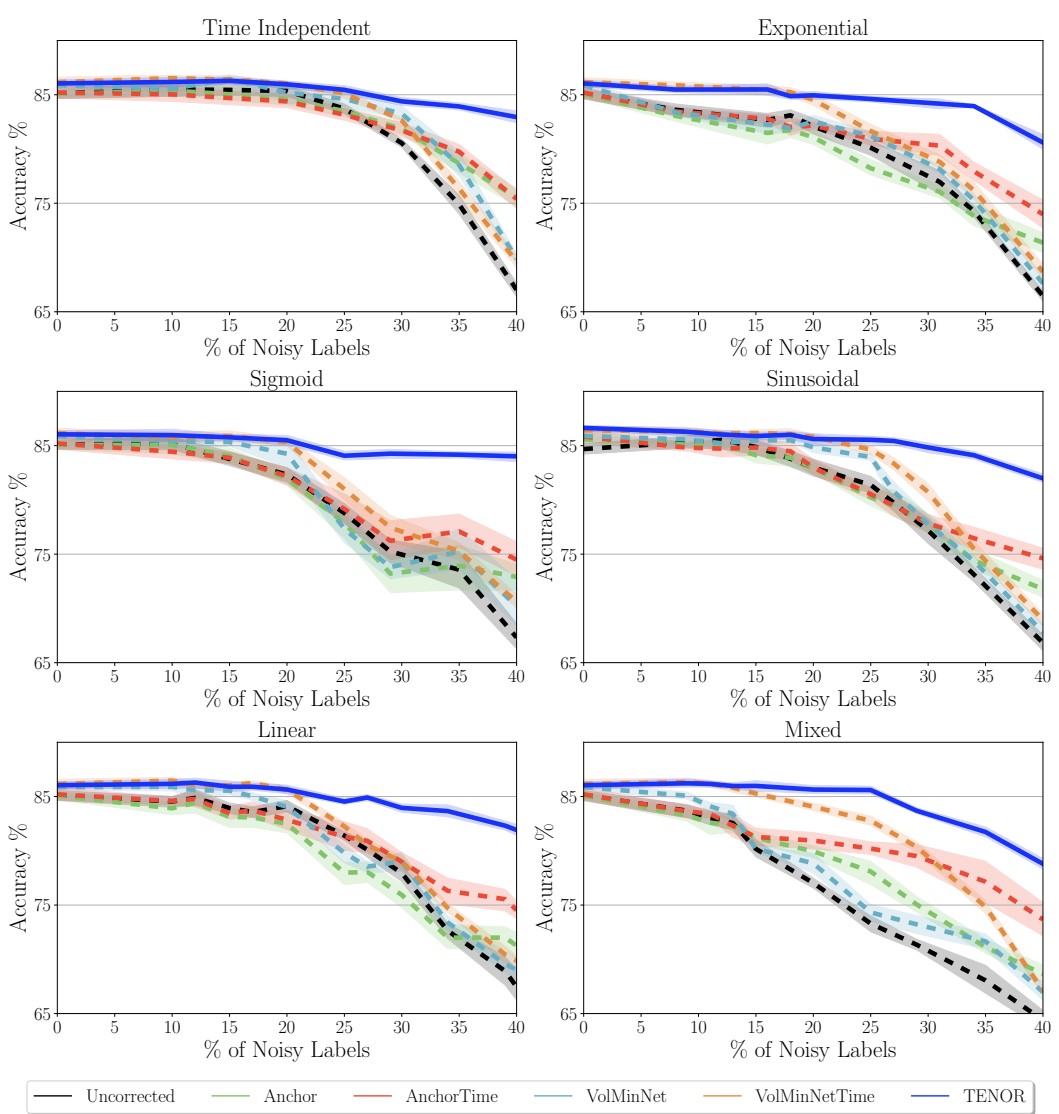

**Figure 12:** Comparison of clean test set Accuracy (%) for `eeg_eye` across varying degrees of temporal label noise comparing all methods. Error bars are st. dev. over 10 runs.

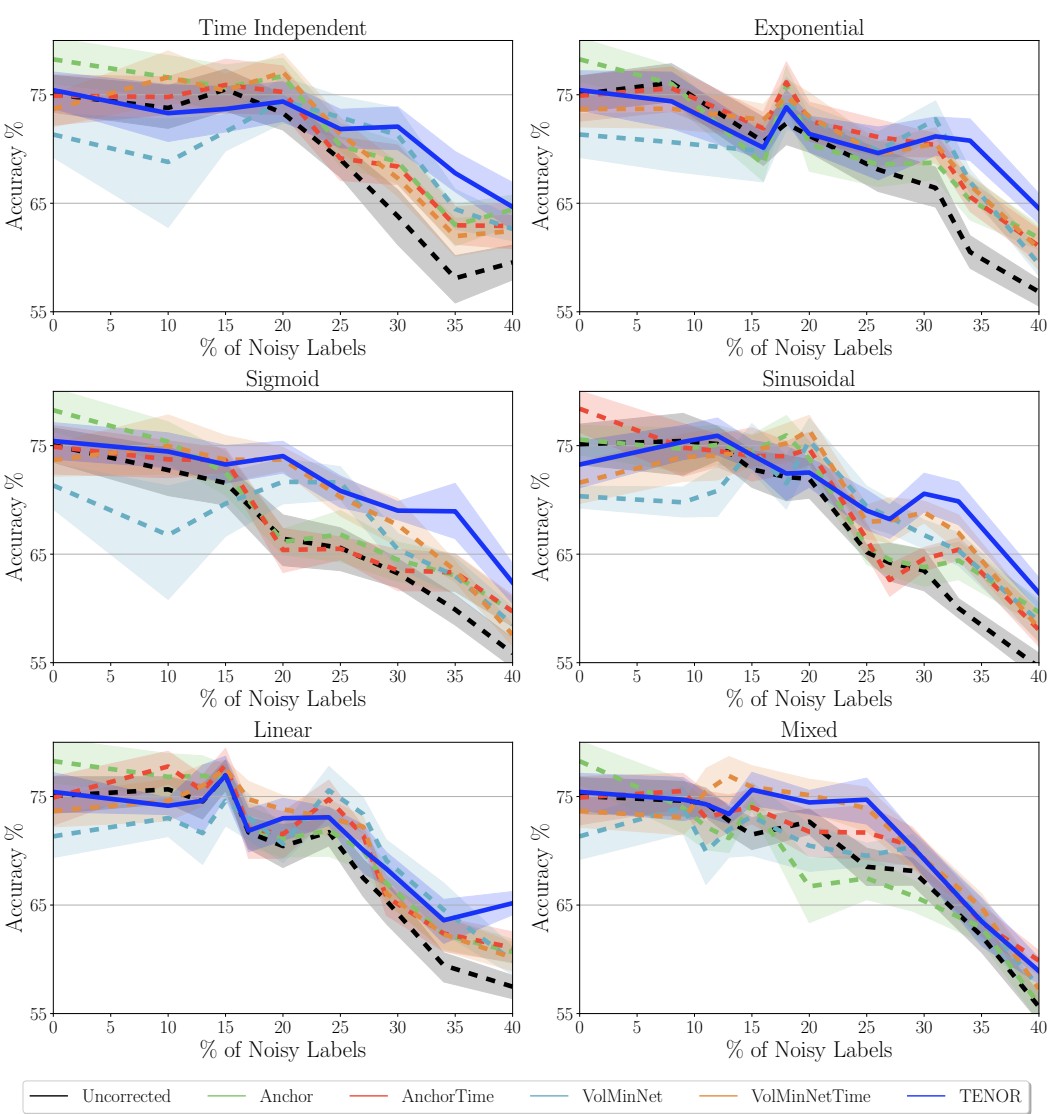

**Figure 13:** Comparison of noisy function reconstruction Mean Absolute Error (MAE) for `synth` across varying degrees of temporal label noise comparing all methods. Error bars are st. dev. over 10 runs.

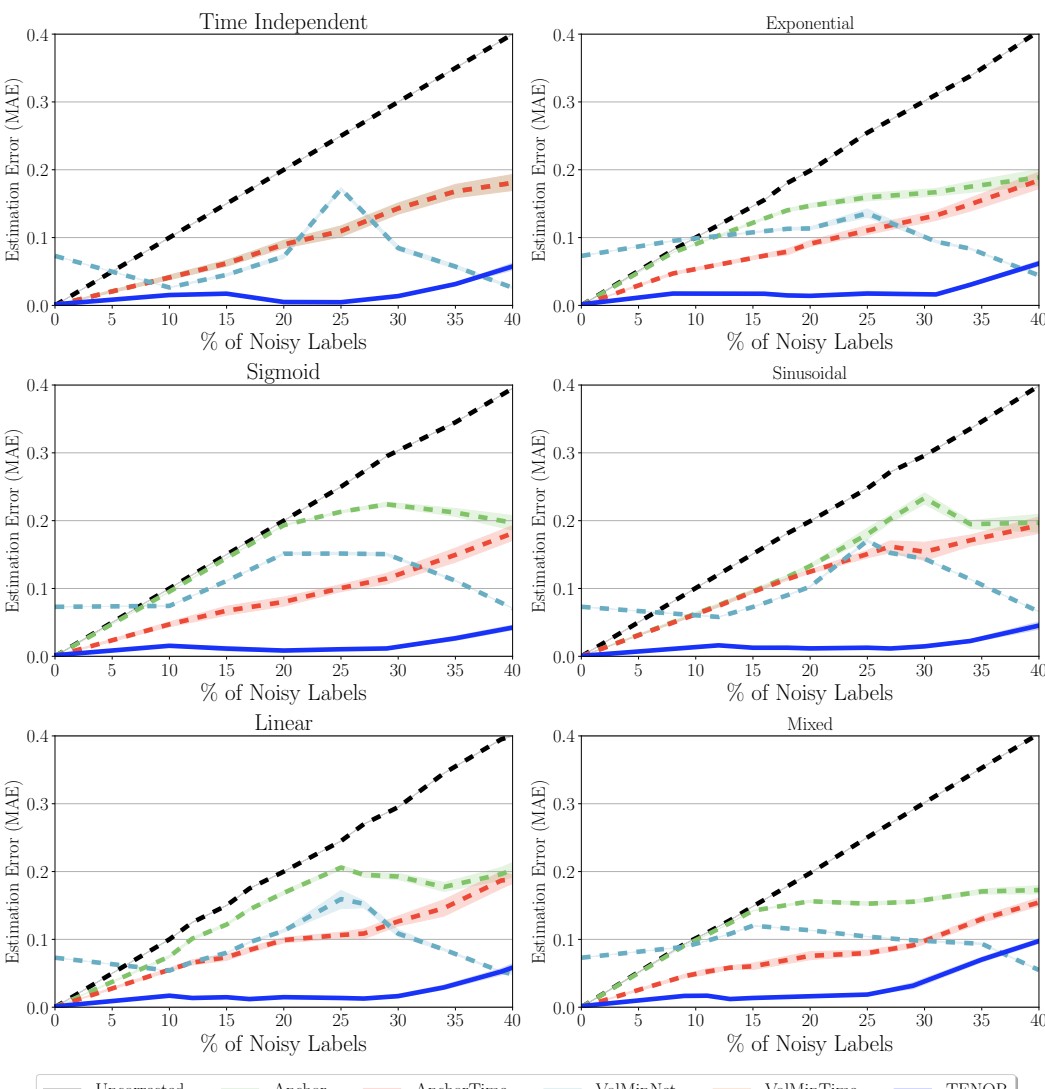

**Figure 14:** Comparison of noisy function reconstruction Mean Absolute Error (MAE) for `har` across varying degrees of temporal label noise comparing all methods. Error bars are st. dev. over 10 runs.

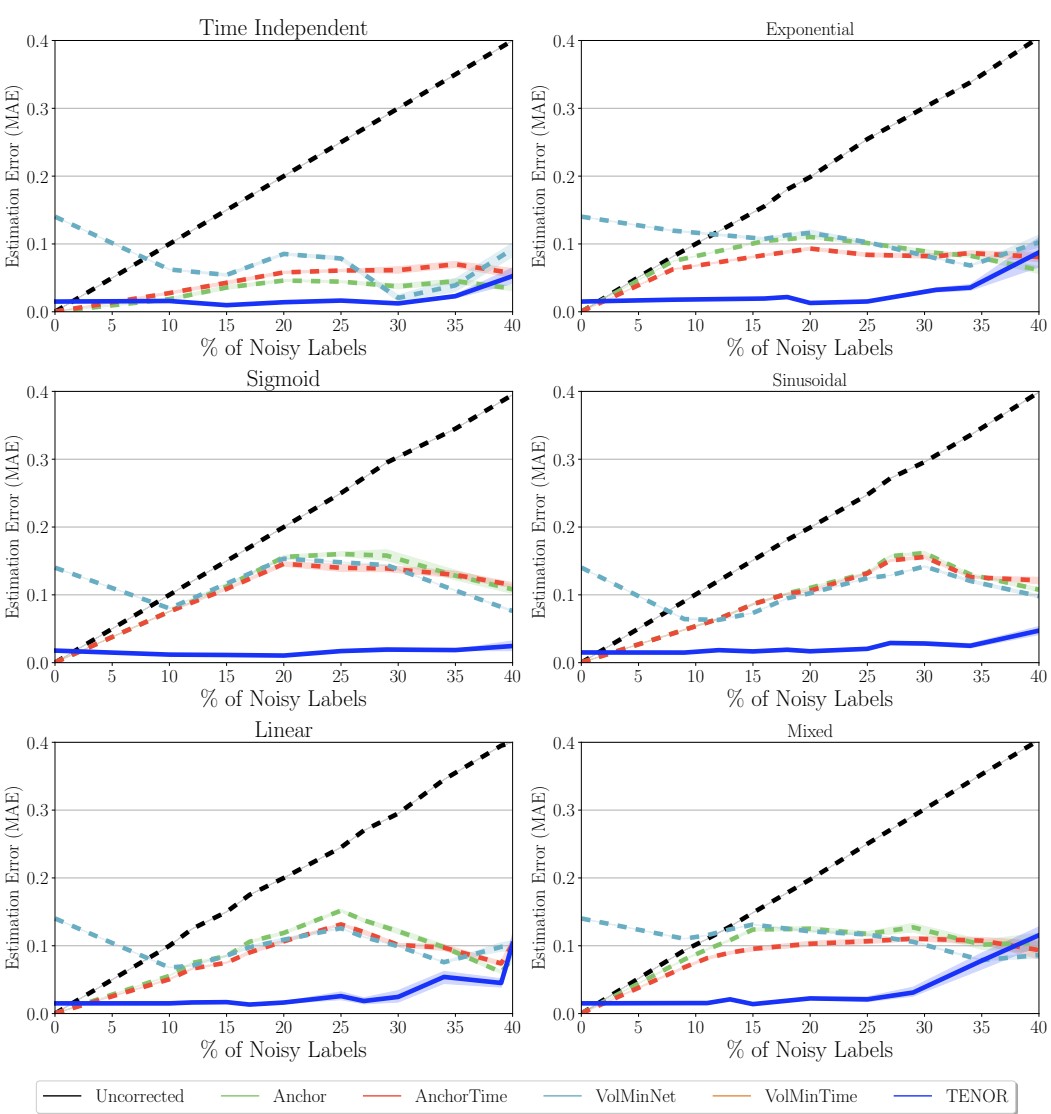

**Figure 15:** Comparison of noisy function reconstruction Mean Absolute Error (MAE) for `har70` across varying degrees of temporal label noise comparing all methods. Error bars are st. dev. over 10 runs.

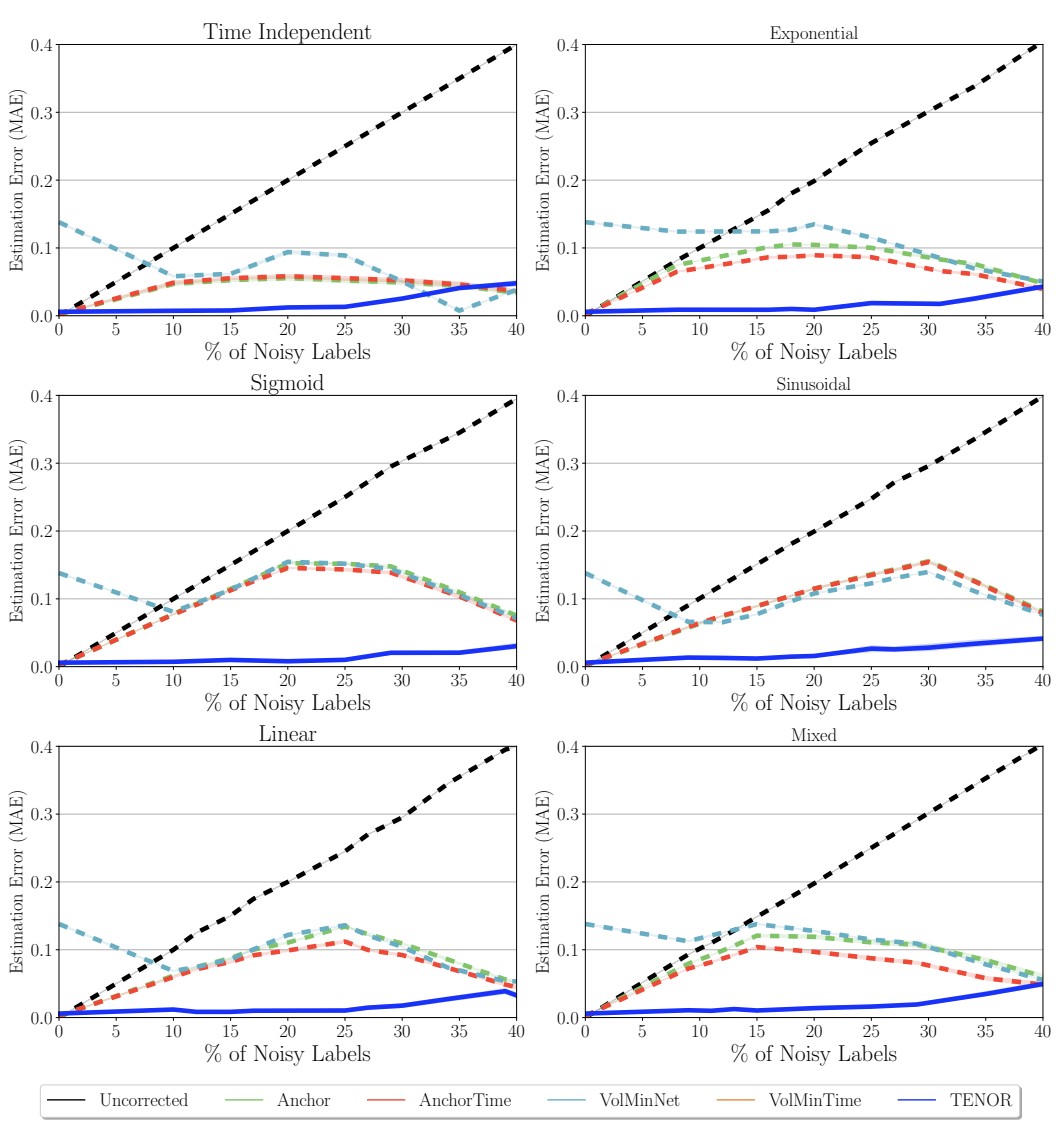

**Figure 16:** Comparison of noisy function reconstruction Mean Absolute Error (MAE) for `eeg_sleep` across varying degrees of temporal label noise comparing all methods. Error bars are st. dev. over 10 runs.

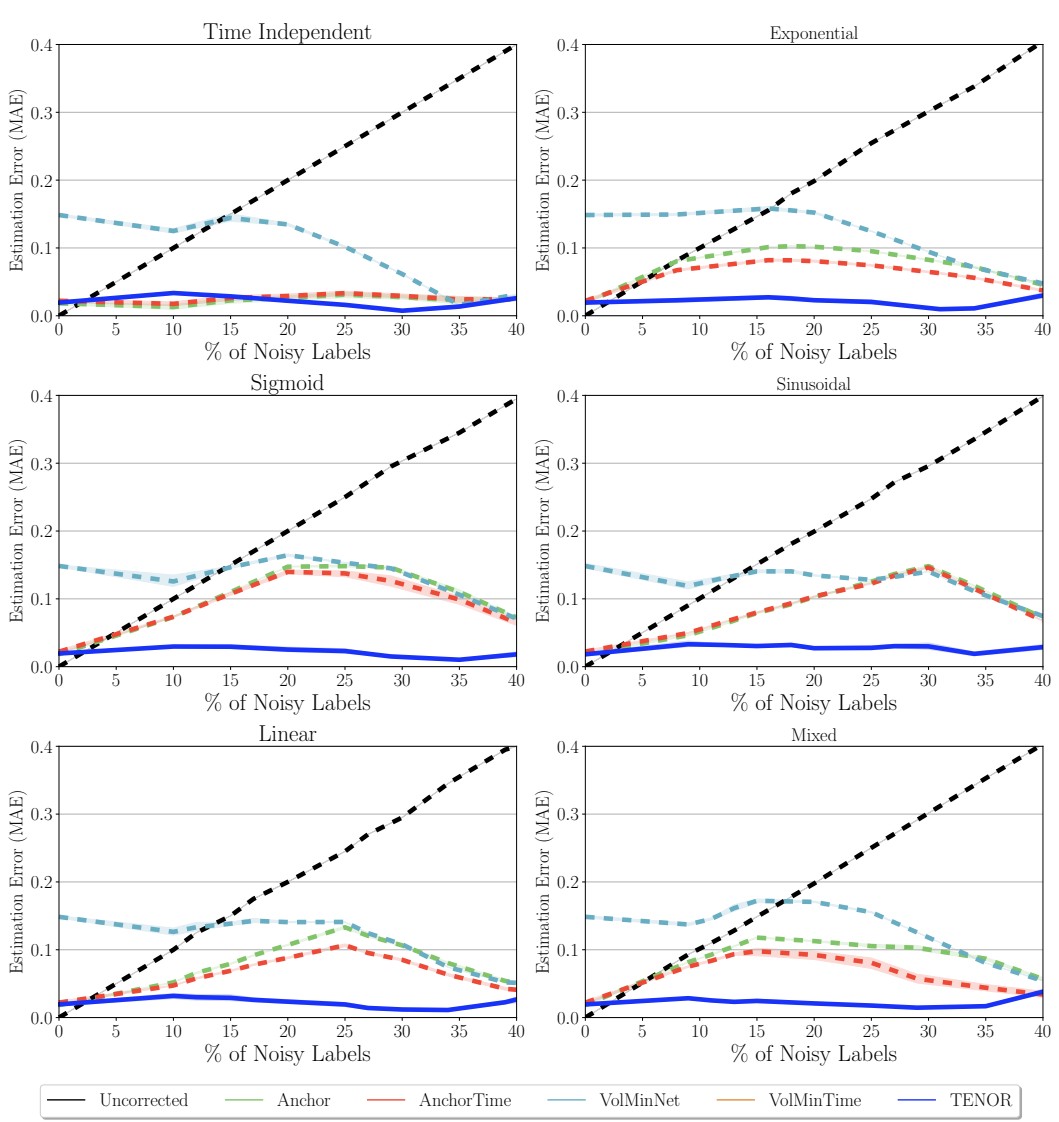

**Figure 17:** Comparison of noisy function reconstruction Mean Absolute Error (MAE) for `eeg_eye` across varying degrees of temporal label noise comparing all methods. Error bars are st. dev. over 10 runs.

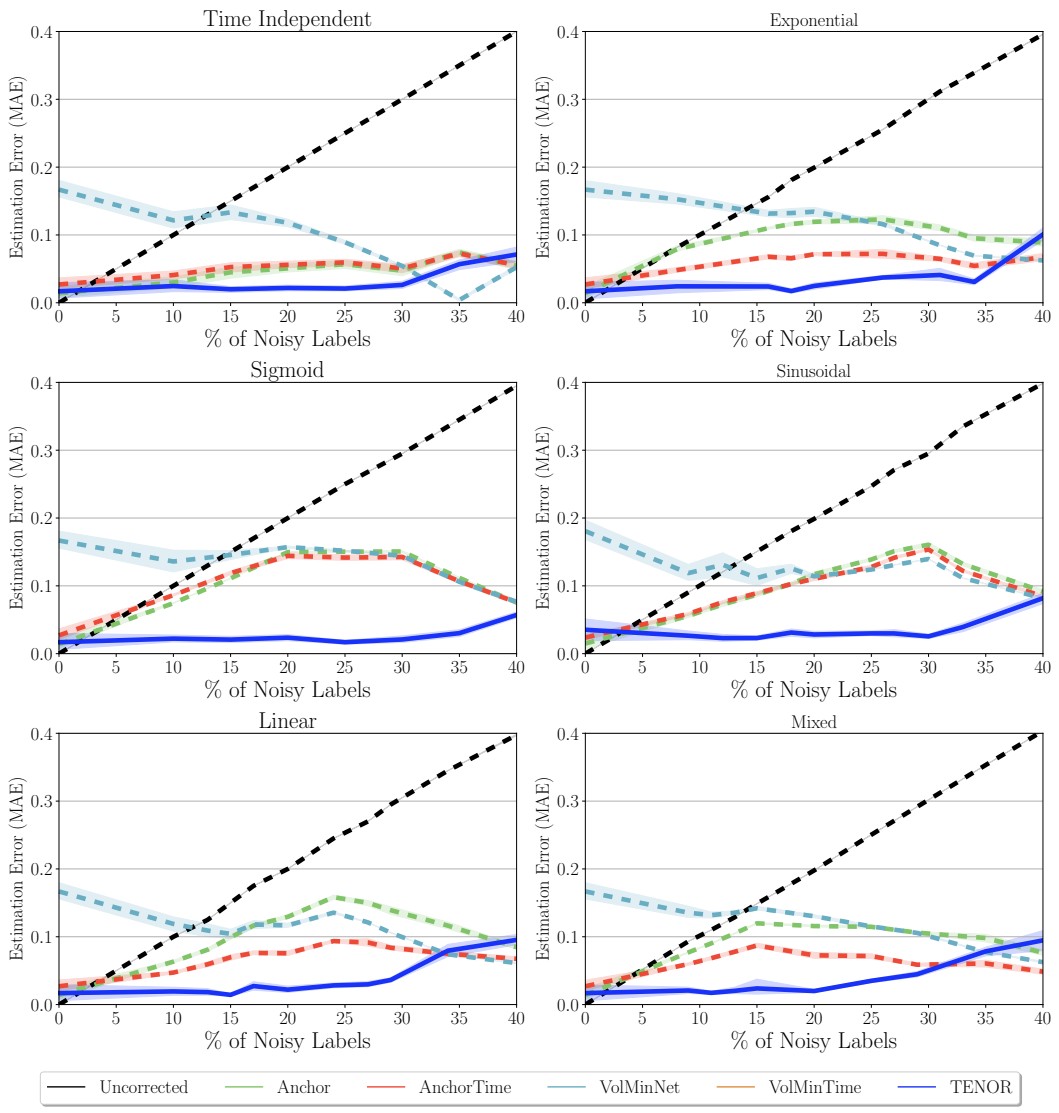

### E.5 Choice of Loss and Optimization Strategy

In our experiments we have slightly different loss terms for TENOR and VolMinTime, with the former penalizing the Frobenius norm and the latter penalizing the LogDet. We also use a special augmentation strategy for TENOR, the augmented Lagrangian method. In order to show that these changes are not what is leading to differences in performance, we compare TENOR to versions of VolMinTime using both types of losses and the augmented Lagrangian optimization method (VolMinTime-AL).

**Figure 18:** Comparison of clean test set Accuracy (%) for `synth` across varying degrees of temporal label noise comparing loss functions for VolMinTime. Error bars are st. dev. over 10 runs.

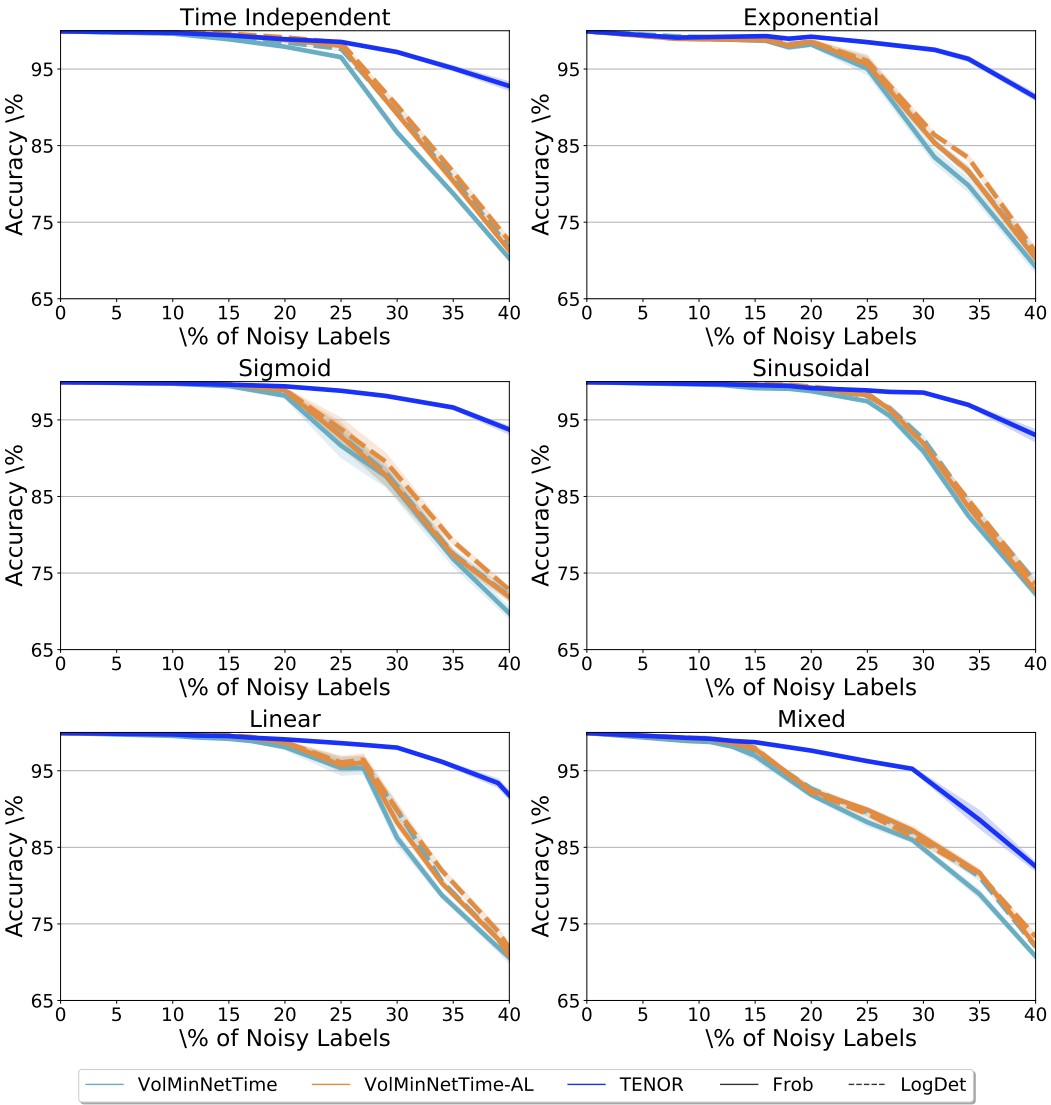

**Figure 19:** Comparison of noisy function reconstruction Mean Absolute Error (MAE) for `synth` across varying degrees of temporal label noise comparing loss functions for VolMinTime. Error bars are st. dev. over 10 runs.

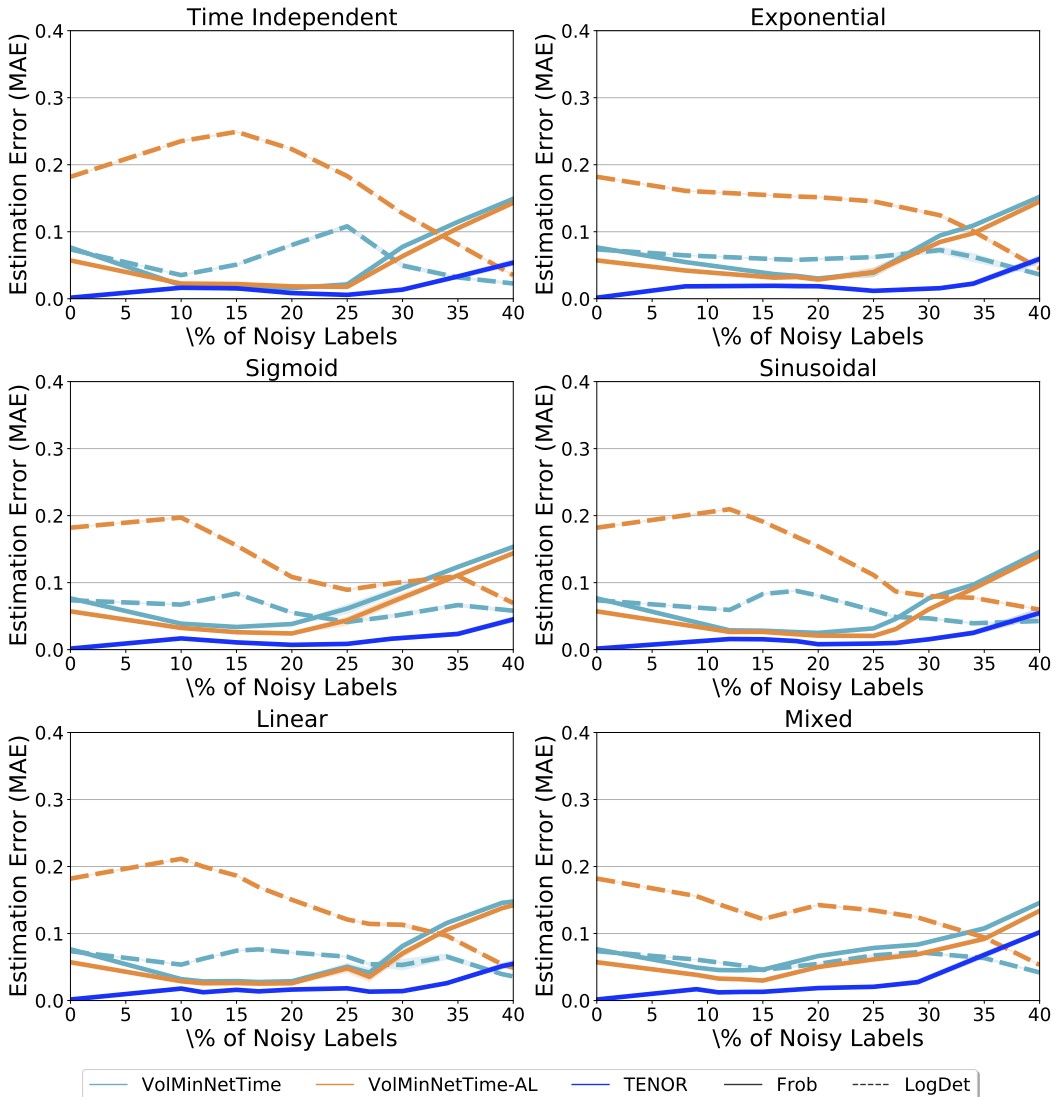

### E.6 CLASS DEPENDENT AND CLASS INDEPENDENT

### E.7 MULTICLASS CLASSIFICATION

**Table 4:** Comparison of clean test-set Accuracy (%) and MAE of all methods on Class Indepenent and Class Dependent sinusoidal label noise for a fixed degree of label noise (30%) on har. Dashed line separates *Static* and *Temporal* methods.

| | | Class Independent | | Class Dependent | |
| --- | --- | --- | --- | --- | --- |
| | | Accuracy ↑ | MAE ↓ | Accuracy ↑ | MAE ↓ |
| Static | Uncorrected | 76.0±5.1 | – | 76.4±3.1 | – |
| | Anchor | 82.0±3.6 | 0.15±0.014 | 84.2±2.2 | 0.13±0.012 |
| | VolMinNet | 86.5±6.0 | 0.13±0.009 | 92.6±1.9 | 0.12±0.012 |
| Temporal | AnchorTime | 81.5±4.3 | 0.14±0.013 | 84.1±2.3 | 0.13±0.010 |
| | VolMinTime | 86.0±5.7 | 0.10±0.015 | 91.5±2.1 | 0.08±0.004 |
| | TENOR | 98.3±0.6 | 0.03±0.005 | 98.4±0.7 | 0.02±0.005 |

**Figure 20:** Comparison of clean test set Accuracy (%) for synth across varying degrees of temporal label noise comparing all methods for 3-class classification. Error bars are st. dev. over 10 runs.

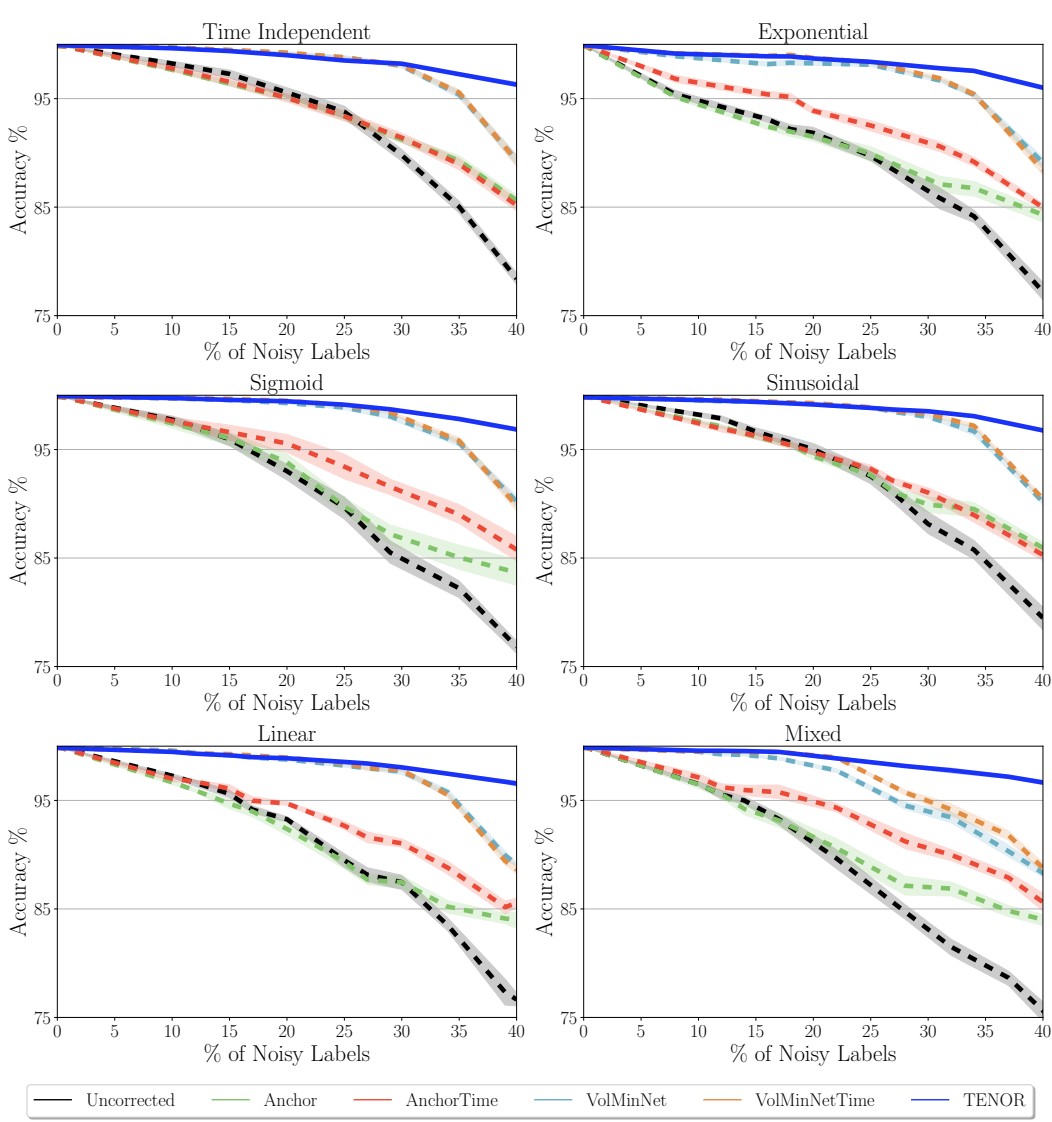

**Figure 21:** Comparison of noisy function reconstruction Mean Absolute Error (MAE) for `synth` across varying degrees of temporal label noise comparing all methods for 3-class classification. Error bars are st. dev. over 10 runs.

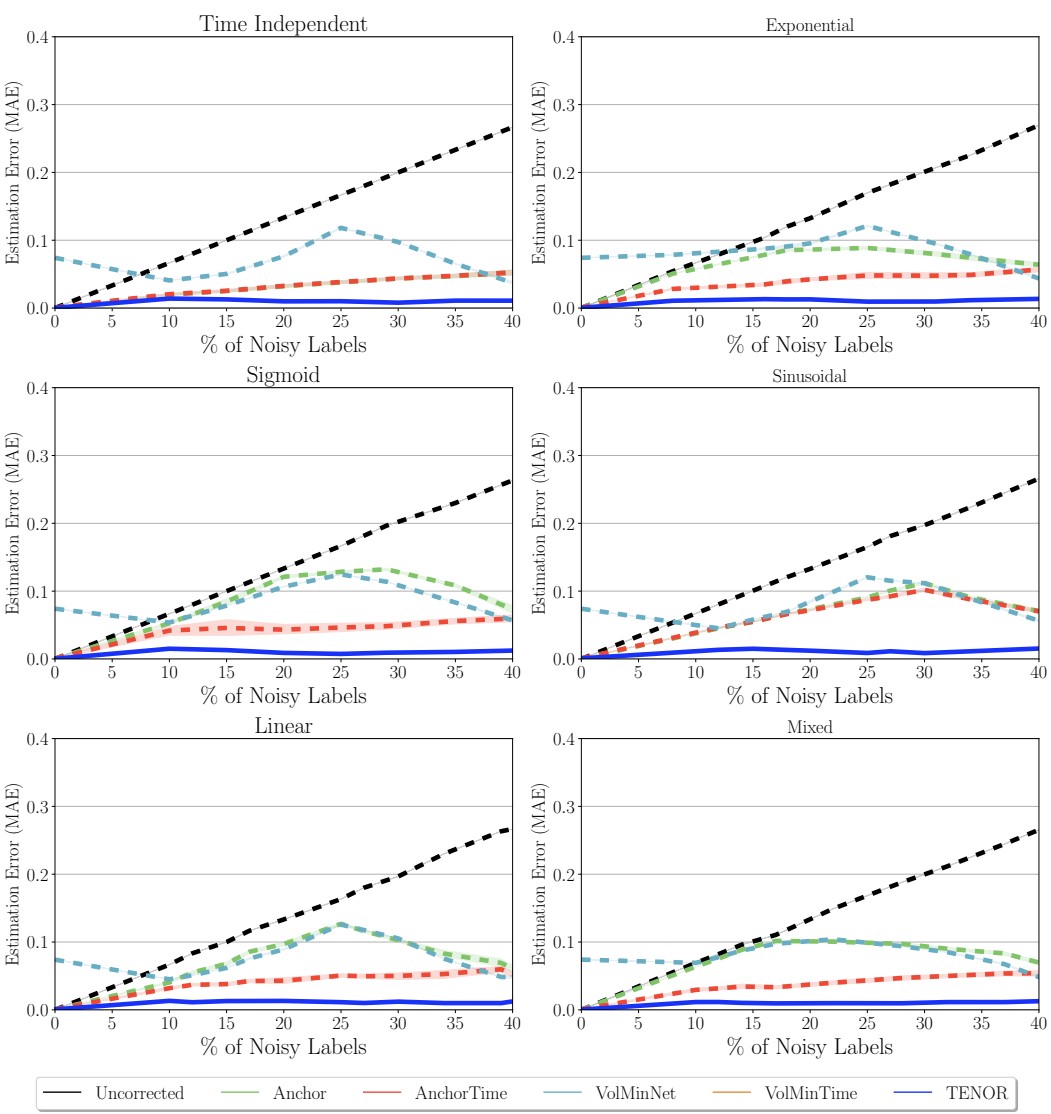

**Figure 22:** Comparison of clean test set Accuracy (%) for `eeg_sleep` across varying degrees of temporal label noise comparing all methods for 3-class classification. Error bars are st. dev. over 10 runs.

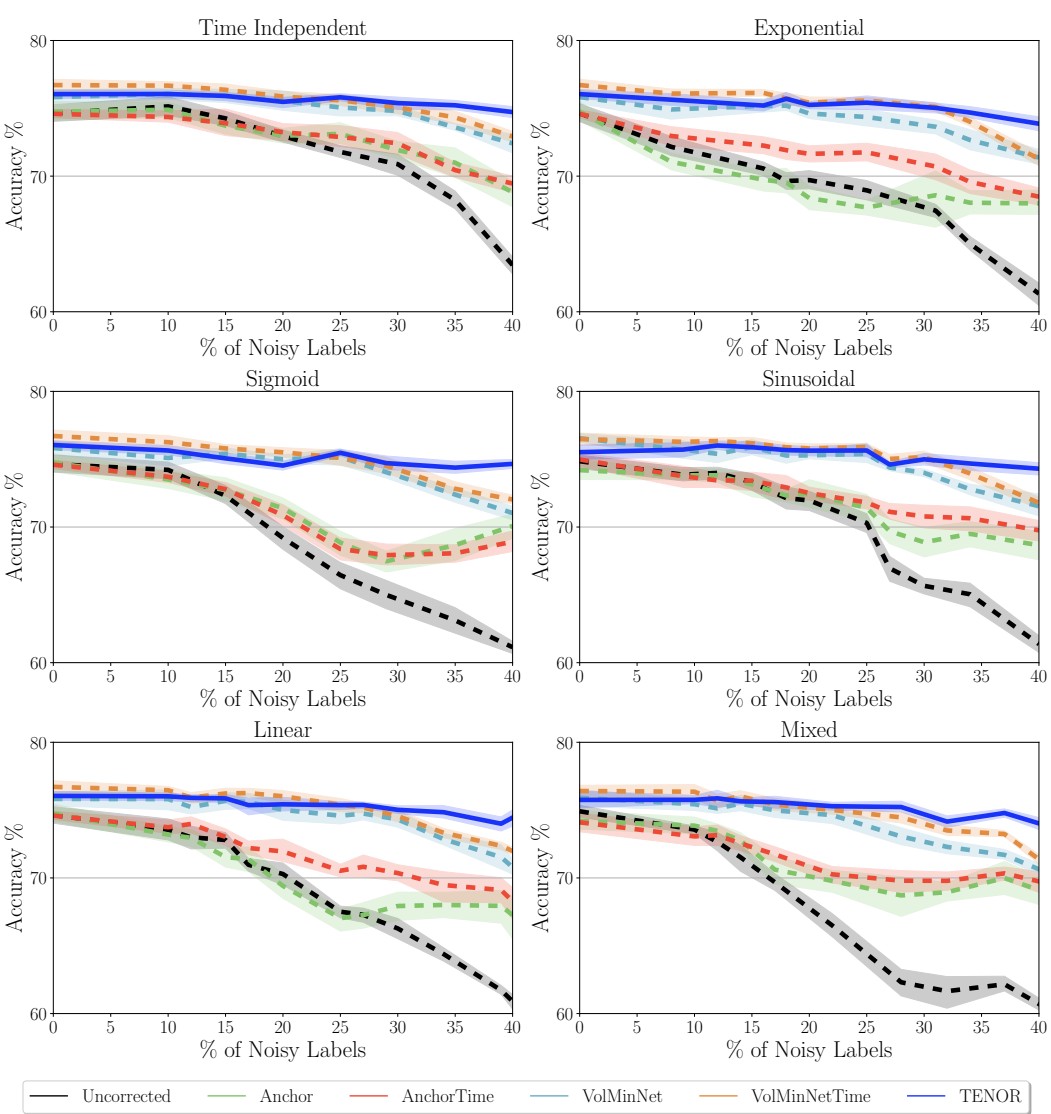

**Figure 23:** Comparison of noisy function reconstruction Mean Absolute Error (MAE) for `eeg_sleep` across varying degrees of temporal label noise comparing all methods for 3-class classification. Error bars are st. dev. over 10 runs.

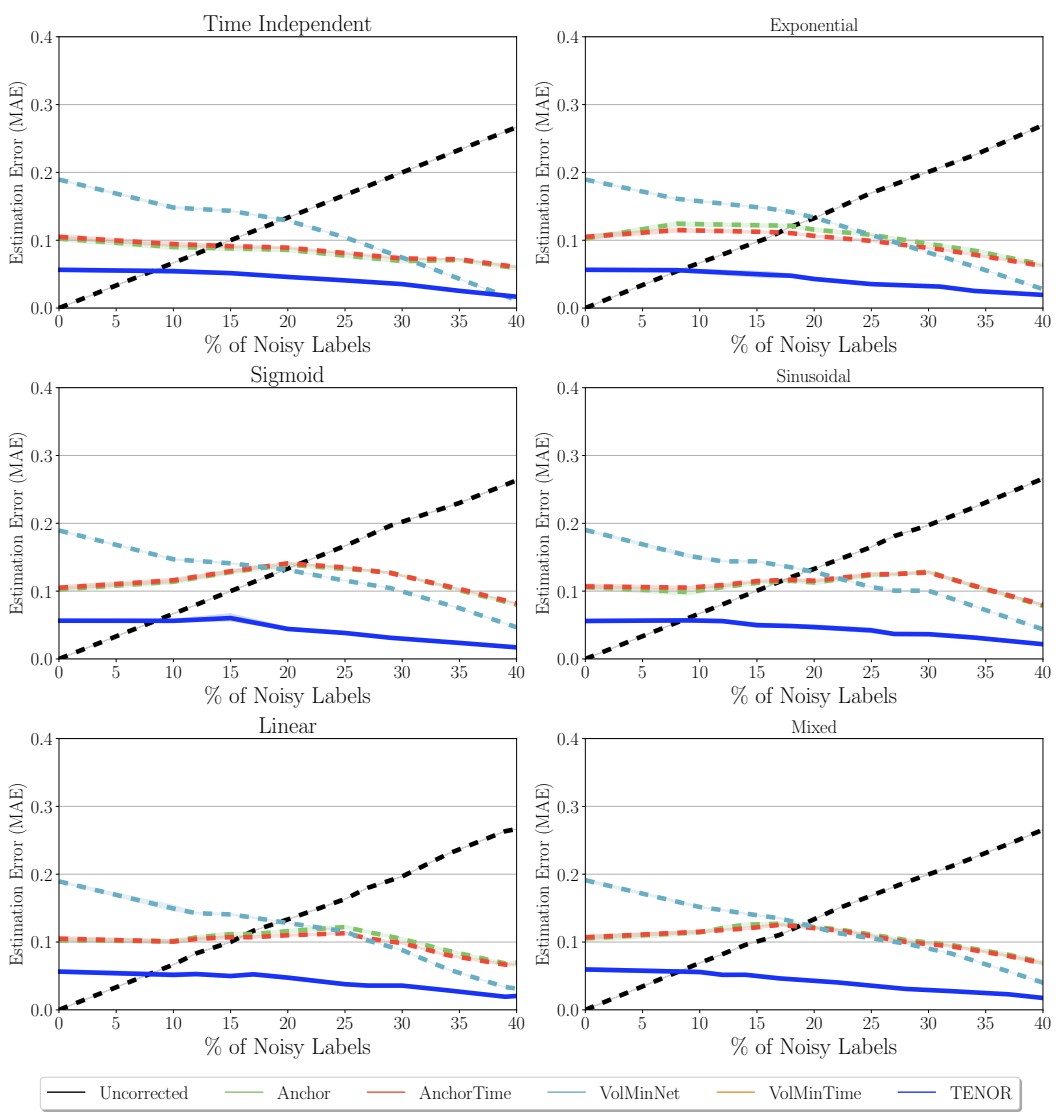

**Figure 24:** Comparison of clean test set Accuracy (%) for `har` across varying degrees of temporal label noise comparing all methods for 4-class classification. Error bars are st. dev. over 10 runs.

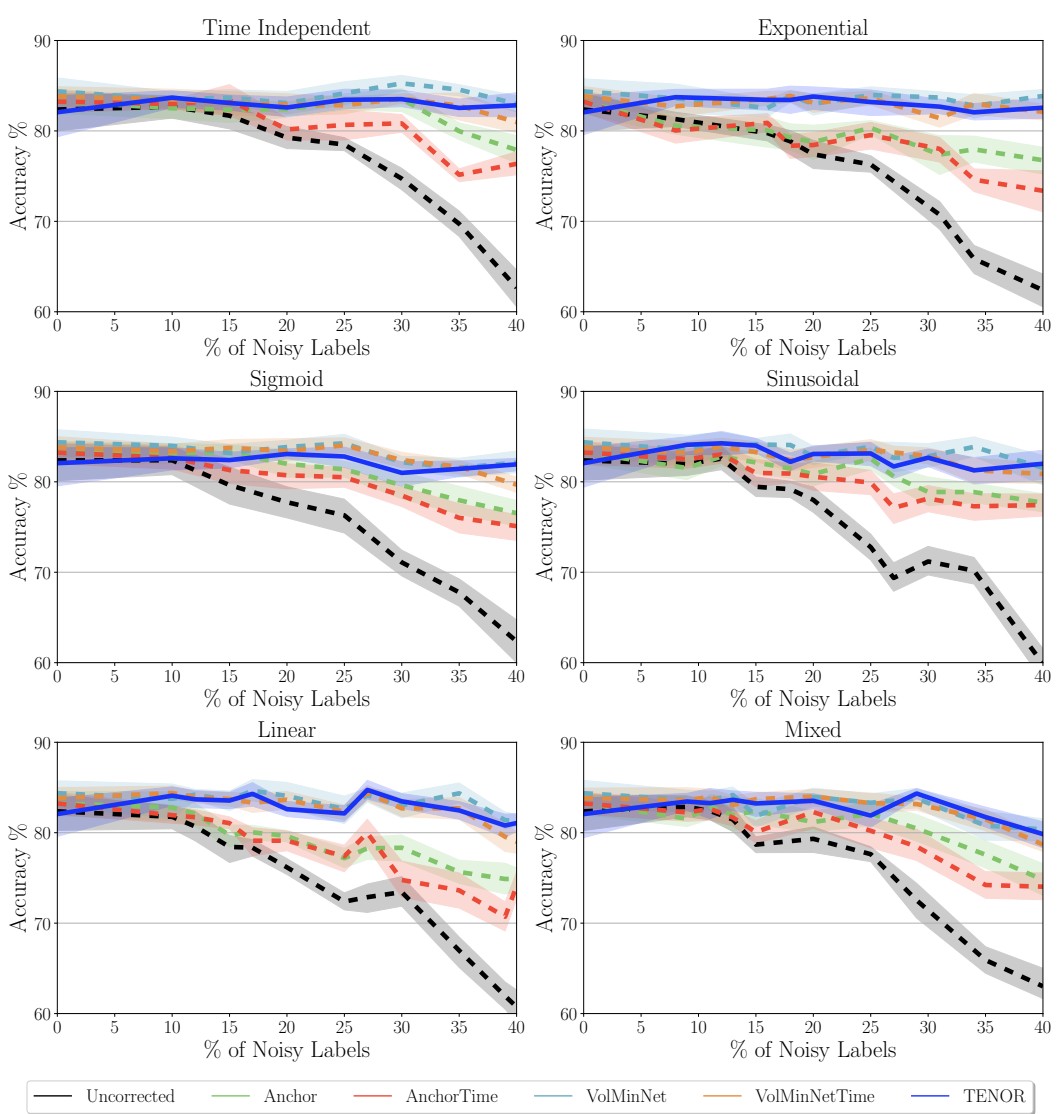

**Figure 25:** Comparison of noisy function reconstruction Mean Absolute Error (MAE) for `har` across varying degrees of temporal label noise comparing all methods for 4-class classification. Error bars are st. dev. over 10 runs.

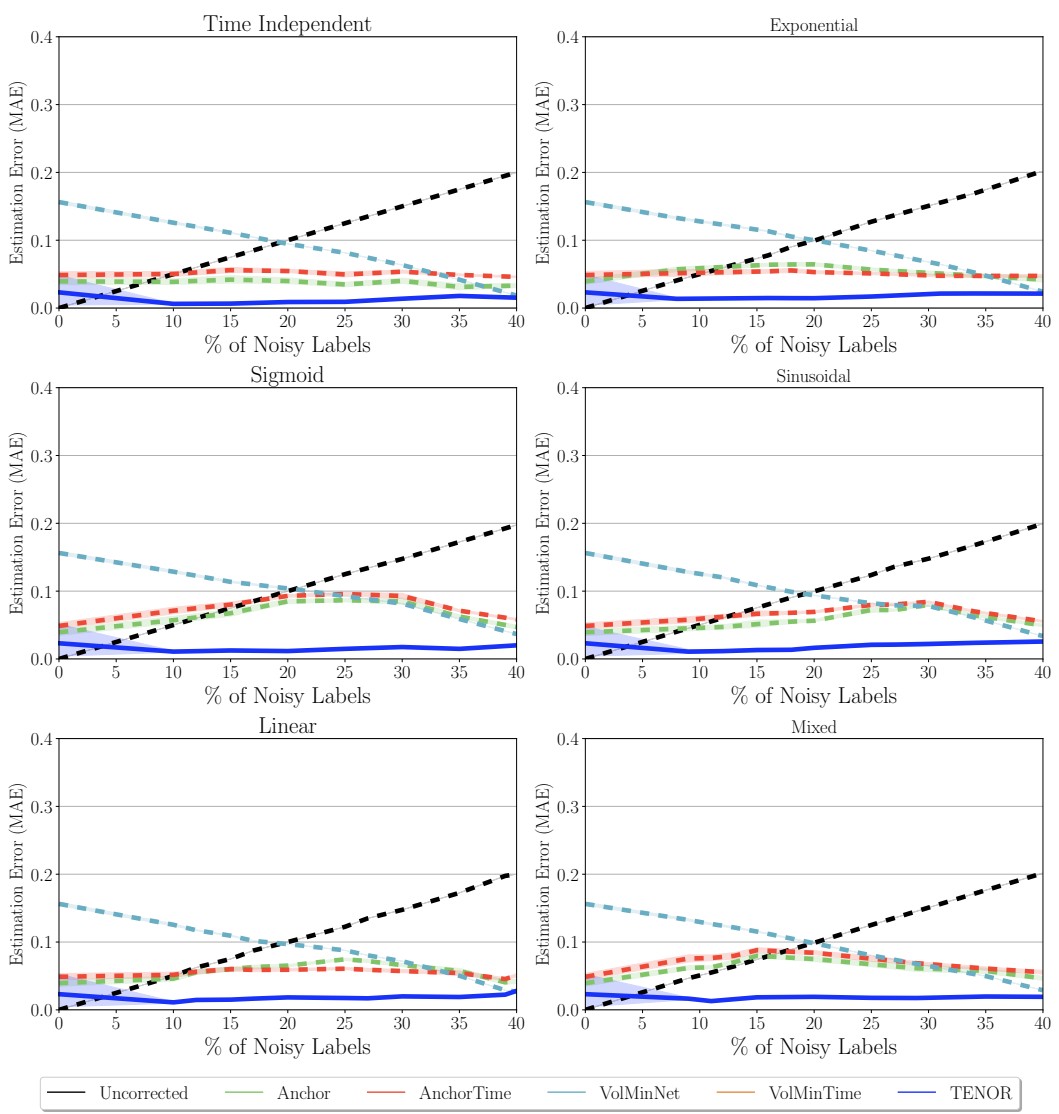

