# OpenReview forum: "Learning with Temporal Label Noise"
_ICLR.cc/2024/Conference — Submitted to ICLR 2024_

### Official Review · Reviewer_ynzB · 2023-10-29

**Soundness:** 3 good
**Presentation:** 3 good
**Contribution:** 3 good
**Rating:** 8
**Confidence:** 4

**Summary:**

Learning from label noise has been widely studied. This paper focuses particularly on the case where noise is introduced over time under specific conditions. This paper introduces the case where noise is time-dependent. Then the authors propose methods that can estimate the temporal label noise function directly from data. The proposed methods improve the performance against existing methods on synthetic and real-world datasets.

**Strengths:**

The paper formally defines a non-trivial noise label setting whose assumption is more realistic than the earlier approach.

The paper addresses empirical improvement on multiple datasets against existing
state-of-the-art algorithms.

The paper is well written and it is easy to follow.

**Weaknesses:**

The paper does not discuss the computational complexity

How about the training time compared to other methods?

Is it possible to show that the datasets satisfy the assumption?

Fix the references. The name of some conferences and journals appears with uppercase letters and others with lowercase letters.

**Questions:**

See Weaknesses Section.

---

> ### Author Response · Authors · 2023-11-17
>
> Thank you for your time and feedback! We address your questions below:
>
> **"`The paper does not discuss the computational complexity. How about the training time compared to other methods?"**
>
> Thanks for raising this point! The compared methods have the same sequential model architecture (GRU), number of gradient updates, and their computational graphs are quite similar. Therefore, differences in complexity will be negligible and arise from the noise-transition estimation function. TENOR is actually significantly more efficient in time and memory complexity than the nearest competing baseline VolMinTime. This efficiency stems from TENOR’s single network to estimate the noise-transition function for all time steps. This process can actually be done in a batched fashion for all time steps at once, with a single gradient update. Meanwhile, VolMinTime learns a separate set of weights at time step, each requiring an independent gradient update. In terms of memory complexity, as the length of the time-series increases, TENOR’s memory cost remains constant as it models for all time. Whereas, VolMinTime stores a separate set of weights at each time, therefore growing linearly with T.
>
> **”Is it possible to show that the datasets satisfy the assumption?"**
>
> Thank you for this question. The assumptions we make regarding modelling sequences are the most common assumptions made by baseline and SOTA models for this problem setup [1-3]. Plus, our findings are the same on synthetic and real-world data, suggesting the assumptions baked into the synthetic data remain valid.
>
> **”Fix the references”**
>
> Thanks for catching this, we have fixed the typos in the references.
>
>
> References
>
> [1] Yoshua Bengio, Réjean Ducharme, and Pascal Vincent. A neural probabilistic language model.
> Advances in Neural Information Processing Systems, 13, 2000.
>
> [2] Yo Joong Choe, Jaehyeok Shin, and Neil Spencer. Probabilistic interpretations of recurrent
> neural networks. Probabilistic Graphical Models, 2017.
>
> [3] Rubanova, Yulia, Ricky TQ Chen, and David K. Duvenaud. "Latent ordinary differential equations for irregularly-sampled time series." Advances in neural information processing systems 32 (2019).

---

### Official Review · Reviewer_Q9Ck · 2023-10-31

**Soundness:** 3 good
**Presentation:** 3 good
**Contribution:** 3 good
**Rating:** 5
**Confidence:** 4

**Summary:**

This paper introduces the concept of a new label noise setting: temporal label noise, where labels are affected by a time-dependent noise function. It presents a method to train a sequential classifier resilient to such noise, directly estimating the temporal label noise function from data. Its main contributions lie in formalizing the temporal label noise problem and devising a novel algorithm to tackle it. Extensive experiments conducted on both synthetic and real-world sequential classification tasks, demonstrate the effectiveness of the proposed method in addressing the temporal label noise.

**Strengths:**

(1) A new label noise setting is introduced which is unexplored before.

(2) A new method named TENOR is proposed to model the temporal label noise.

(3) Extensive experimental results demonstrated the effectiveness of the proposed method in addressing the temporal label noise.

**Weaknesses:**

The paper delves into an interesting temporal label noise learning problem, yet it contains several aspects that could benefit from improvement. While the proposed method introduces a fresh label noise setting, it lacks clarity regarding the motivation behind formalizing this new paradigm. Could the temporal label noise setting serve as a replacement for existing label noise settings? Offering reasonable explanations for these concerns can enhance the significance of the new settings.

**Questions:**

(1) What motivate you introduce this new label noise setting? Does the temporal label noise exist in the real-world application?

(2) The authors assume that ‘the sequence of noisy labels is independent of the features given the true labels’ can be strong, since the label noise in the real-world can be instance depended and class depended.

(3) The comparison methods can be outdated. It is better to consider more SOTA methods as baseline. Furthermore, how you adaptive the baseline to fit your specific setting.

(4) Is the proposed method only applicable to binary classification? It is advisable to adapt the real-world noisy label learning dataset Clothing1M to your framework to validate the efficacy of the proposed method.

---

> ### Author Response · Authors · 2023-11-17
>
> Thank you for your time and feedback! We respond to your questions below:
>
> **"What motivated you to introduce this new label noise setting? Does the temporal label noise exist in the real-world?"**
>
> Thank you for asking this question! We were actually motivated to work on this because of real-world applications that we work on in the healthcare domain. In particular, user-annotated human activity time series data, where participants notoriously mislabel their activities depending on when and what they are doing.
>
> Stepping back, we identified that this problem arises in a substantial class of real-world applications where labels are collected over time, especially when people label their own data over time. Well-known examples include:
>
> 1) **Human activity recognition**: Human activity recognition models detect people’s activities over time, usually from wearable devices or smartphones. For example, smartphone apps may tailor notifications to someone’s behaviour. But training labels are provided by people annotating their own activities over time [1,2]. Such self-annotation is notoriously noisy. For example, people often inaccurately report when they fall asleep or wake up [3]. Such annotations suffer from temporal label noise because some times have more noise than others (e.g., around sleep).
> 2) **Modelling mental health over time**: Consider suicide-risk prediction tools that use wearable sensors to alert doctors of patient risks. These models are trained on self-reported indicators of suicide-risk [4]. But studies consistently show that users can fear repercussions and underrepresent their true suicide risk [5]. In this case, periods of time where individuals are suicidal are subject to increased label noise.
> 3) **Temporal bias in grading**: When grading student assignments, later assignments can receive worse grades, likely as graders tire [6]. This sequential bias implies label noise increased over time in the 30 million records studied in this work. We have included a new citation to this case in the Introduction.
>
> **"The authors assumption that ‘the sequence of noisy labels is independent of the features given the true labels’ can be strong, since the label noise in the real-world can be instance-dependent and class-dependent."**
>
> You are right that this may be a strong assumption. We wanted to be explicit about this assumption in the manuscript.
>
> You are right that there are other variants of real-world label noise that can arise. Stepping back, we actually wanted to focus on the canonical setting in which we had class-conditional label noise that is feature-independent – this was important to pinpoint how methods and performance would change as a result of temporal effects.
>
> Our method is built to handle class-dependent label noise. Handling instance-dependent is out of scope (and generally a challenging problem even in the non-temporal, static setting)  [7-9].
>
> We also consider the setting where noise is simultaneously class- and time-dependent (that is, a unique time-dependent noise process for each class). The temporal noise type “Mixed” reported in the paper represents class- and time-dependent noise. We showcase the superior performance of TENOR in this case (Table 1)! Additionally, we have also added experiments on synthetic and real-world multiclass sequential classification tasks, showing that TENOR has strong performance in these settings as well (Appendix E.7). We believe this further increases the utility and generalizability of our work.
>
> **"The comparison methods can be outdated. It is better to consider more SOTA methods as baseline. Furthermore, how you adaptive the baseline to fit your specific setting"**
>
> We tried to choose baselines that were used in prior work:  VolMinNet (ICML 2021) [10] and Anchor (CVPR 2017) [11]. We’re happy to run experiments against other methods if you would like (just let us know). In general, we think that there is a marginal benefit in adding more baseline methods since they assume that label noise is static over time and will fail when the label noise is temporal (albeit at different rates). We show that such static methods are insufficient, even if they estimate the noise perfectly (Fig 2), so more-extensive comparisons are out of scope. As it stands, these points apply to other methods and future attempts to make existing methods time-dependent.
>
> One point that we would like to make is that some of the “baseline methods” that we present in the experiments are actually methods that we developed ourselves (e.g.: VolMinTime and AnchorTime)! These are methods where we take the existing static method and modify it to a temporal setting as described in Section 4.4 . The results show that some of the issues resulting from temporal label noise can be mitigated by allowing for temporal variation. Our results show that this improves their performance, though our approach remains superior.

---

> ### Author Response · Authors · 2023-11-17
>
> **"It is advisable to adapt the real-world noisy label learning dataset Clothing1M to your framework to validate the efficacy of the proposed method"**
>
> Thanks for mentioning this. We considered applying this dataset since it contains real label noise. Unfortunately, Clothing1M is not naturally applicable to our setting since each instance only has a single label – i.e., there is no “sequence” of labels over time.
>
> Given your suggestion for real-world noisy labels, we did run experiments  with the “ExtraSensory” dataset [12] which also contains “real world noisy labels” for a time series prediction task.  ExtraSensory includes human activity data from smartphones and smartwatches from 60 users over a timespan of 300K minutes.
>
> In this case, we then train binary classifiers to predict when users are asleep or awake, which is a key benchmark task. In contrast to HAR and HAR70 datasets in our experiments, the labels in ExtraSensory are user-provided and subject to label noise (since users often misreport falling asleep and waking up, especially when they are sleepy or in a rush to leave in the morning).
>
> Our results for this dataset show that TENOR successfully identifies an interpretable temporal noise function (Appendix E.2) and that TENOR predicts there exists higher label noise near sleep/awake transitions. We also see that TENOR is responsive to these temporal effects. In this case, we cannot validate if these are the correct predictions since we do not have any cleaned ground truth labels (like HAR, HAR70, or even Clothing1M), but this is the kind of behaviour we would expect to see from a good model in this setting. We included these results in Appendix E.2 and will work to integrate them into the Experiments if you think this is valuable.
>
> **"Is the proposed method only applicable to binary classification?"**
>
> Our proposed method works for multiclass classification problems! We have included results from new experiments on multiclass data for synthetic and real-world tasks (ranging from 2 to 4 classes). We find that our method still outperforms existing methods under a variety of temporal noise functions. Please refer to the following table, demonstrating performance on 3-class classification for the synthetic dataset. Complete results can be found in Appendix E.7.
>
> |             | Time Independent | Time Independent | Exponential  | Exponential    | Linear       | Linear         | Sigmoidal    | Sigmoidal      | Sinusoidal   | Sinusoidal     | Mixed        | Mixed          |
> |-------------|------------------|------------------|--------------|----------------|--------------|----------------|--------------|----------------|--------------|----------------|--------------|----------------|
> |             | *Accuracy*         | *MAE*              | *Accuracy*     | *MAE*           | *Accuracy*     | *MAE*            | *Accuracy*     | *MAE*            | *Accuracy*     | *MAE*            | *Accuracy*     | *MAE*            |
> | Uncorrected | 81.7$\pm$3.4     | NA            | 77.2$\pm$1.3 | NA         | 77.0$\pm$1.6 | NA         | 79.5$\pm$2.9 | NA          | 79.5$\pm$1.7 | NA          | 76.6$\pm$2.4 | NA          |
> | Anchor      | 87.5$\pm$2.0     | 0.05$\pm$0.006   | 84.3$\pm$0.9 | 0.06$\pm$0.005 | 84.1$\pm$1.1 | 0.06$\pm$0.009 | 84.4$\pm$2.0 | 0.09$\pm$0.019 | 85.9$\pm$0.9 | 0.07$\pm$0.004 | 84.3$\pm$1.0 | 0.07$\pm$0.010 |
> | VolMinNet   | 92.4$\pm$3.1     | 0.05$\pm$0.014   | 89.1$\pm$0.8 | 0.04$\pm$0.002 | 89.5$\pm$0.6 | 0.05$\pm$0.003 | 92.9$\pm$2.8 | 0.07$\pm$0.013 | 90.2$\pm$0.4 | 0.06$\pm$0.003 | 89.0$\pm$1.5 | 0.05$\pm$0.013 |
> | AnchorTime  | 87.1$\pm$2.1     | 0.05$\pm$0.006   | 85.0$\pm$0.5 | 0.06$\pm$0.005 | 85.4$\pm$0.9 | 0.06$\pm$0.007 | 87.4$\pm$2.3 | 0.06$\pm$0.008 | 85.3$\pm$0.8 | 0.07$\pm$0.004 | 86.4$\pm$2.0 | 0.05$\pm$0.006 |
> | VolMinTime  | 90.0$\pm$3.3     | 0.03$\pm$0.008   | 86.8$\pm$1.0 | 0.05$\pm$0.001 | 87.0$\pm$1.3 | 0.04$\pm$0.002 | 90.9$\pm$3.3 | 0.04$\pm$0.007 | 88.4$\pm$0.8 | 0.05$\pm$0.001 | 87.8$\pm$2.1 | 0.04$\pm$0.004 |
> | TENOR       | **96.8$\pm$0.6**     | **0.01$\pm$0.002**   | **96.0$\pm$0.3** | **0.01$\pm$0.002** | **96.6$\pm$0.4** | **0.01$\pm$0.002** | **97.3$\pm$0.6** | **0.01$\pm$0.002** | **96.8$\pm$0.3** | **0.02$\pm$0.002** | **96.8$\pm$0.5** | **0.01$\pm$0.002** |

---

> > ### Author Response · Authors · 2023-11-17
> >
> > References
> >
> > [1] Parde, Natalie, and Rodney Nielsen. "Finding patterns in noisy crowds: Regression-based annotation aggregation for crowdsourced data." Proceedings of the 2017 Conference on Empirical Methods in Natural Language Processing. 2017.
> >
> > [2] Kwon, Hyeokhyen, Gregory D. Abowd, and Thomas Plötz. "Handling annotation uncertainty in human activity recognition." Proceedings of the 2019 ACM International Symposium on Wearable Computers. 2019.
> >
> > [3] Lauderdale, Diane S., et al. "Self-reported and measured sleep duration: how similar are they?." Epidemiology (2008): 838-845.
> >
> > [4] Whiting, Daniel, and Seena Fazel. "How accurate are suicide risk prediction models? Asking the right questions for clinical practice." BMJ Ment Health 22.3 (2019): 125-128.
> >
> > [5] Hom, Melanie A., et al. "“Are you having thoughts of suicide?” Examining experiences with disclosing and denying suicidal ideation." Journal of Clinical Psychology 73.10 (2017): 1382-1392.
> >
> > [6] Pei, Jiaxin, Zhihan Helen Wang, and Jun Li. "30 Million Canvas Grading Records Reveal Widespread Sequential Bias and System-Induced Surname Initial Disparity." (2023).
> >
> > [7] Liu, Sheng, et al. "Robust training under label noise by over-parameterization." International Conference on Machine Learning. PMLR, 2022.
> >
> > [8] Zhu, Zhaowei, Jialu Wang, and Yang Liu. "Beyond images: Label noise transition matrix estimation for tasks with lower-quality features." International Conference on Machine Learning. PMLR, 2022.
> >
> > [9] Yong, L. I. N., et al. "A Holistic View of Label Noise Transition Matrix in Deep Learning and Beyond." The Eleventh International Conference on Learning Representations. 2022.
> >
> > [10] Li, Xuefeng, et al. "Provably end-to-end label-noise learning without anchor points." International conference on machine learning. PMLR, 2021.
> >
> > [11] Patrini, Giorgio, et al. "Making deep neural networks robust to label noise: A loss correction approach." Proceedings of the IEEE conference on computer vision and pattern recognition. 2017.
> >
> > [12] Vaizman, Yonatan, Katherine Ellis, and Gert Lanckriet. "Recognizing detailed human context in the wild from smartphones and smartwatches." IEEE pervasive computing 16.4 (2017).

---

### Official Review · Reviewer_PKMs · 2023-11-04

**Soundness:** 2 fair
**Presentation:** 3 good
**Contribution:** 2 fair
**Rating:** 3
**Confidence:** 4

**Summary:**

This paper studies the problem of learning with label noise under temporal classification, specifically when label noise could be generated from a non-i.i.d function, and existing works could easily fail. To tackle the problem of temporal label noise, the authors first demonstrate that under certain settings, applying loss correction to temporal label noise can obtain a risk-consistent classifier defined over a noise-free distribution. Based on this principle, the authors further proposed TENOR, a noise transition estimation method for temporal classification.

**Strengths:**

1. This paper proposed a meaningful and relatively under-studied question - how to tackle sequentially correlated label noise.

2. The authors attempt to address this issue from a theoretical prospective, by proving that the loss correction framework under the i.i.d setting can be applied to temporal classification.

3. In addition to the end-to-end learning framework and volume-minimization from existing work [1], authors further proposed a regularization term that enforces the off-diagonal value of the transition matrix to not vanish, which could be useful when the noise rate is high.

[1] Li, Xuefeng, et al. "Provably end-to-end label-noise learning without anchor points." International conference on machine learning. PMLR, 2021.

**Weaknesses:**

$\textbf{Major issues:}$

1. Simply assuming Assumption 2-4 ignores the main technical challenge in applying loss correction to sequential data - this bypasses the significant challenges such as how the corrected loss at each time step correlates with other time steps.

2. The noise transition estimator is directly borrowed from existing works, however, there is an in-depth discussion on how these methods are suitable for sequential scenarios.

$\textbf{Minor issues:}$

1. Some strong assumptions are assumed, namely, Assumptions 2-4. The authors did not discuss how those assumptions are valid in a real-world setting, and intuitively speaking, those assumptions can easily be violated.

2. If we penalize the estimated transition matrix from the identity matrix, in a low noise setting, the transition matrix will tend to produce underconfident loss, hence impairing the training of TENOR, which can be observed in Figure 3.

**Questions:**

1. Named entity recognition (NER) with label noise is a highly related domain (both dealing with sequentially correlated label noise), some existing works have already been proposed in this area [1,2], and authors should discuss the similarities and differences between those domains, and if possible, apply TENOR to NER to better showcase the performance superiority.

$\textbf{General improvement advises for authors:}$

1. Authors should consider more mild assumptions that enable loss correction to be applied to temporal classification.

2. The proposed TENOR framework is significantly overlapped with the existing method, authors should devise new methods that are specifically designed for temporal classification (orientated to more specific technical challenges).

[1] Liu, Kun, et al. "Noisy-Labeled NER with Confidence Estimation." Proceedings of the 2021 Conference of the North American Chapter of the Association for Computational Linguistics: Human Language Technologies. 2021.

[2] Huang, Xiusheng, et al. "Named entity recognition via noise aware training mechanism with data filter." Findings of the Association for Computational Linguistics: ACL-IJCNLP 2021. 2021.

---

> ### Author Response · Authors · 2023-11-17
>
> Thank you for your time and feedback! We address your questions below:
>
> **"Named entity recognition (NER) with label noise is a highly related domain... authors should discuss the similarities and differences between those domains, and if possible, apply TENOR to NER to better showcase the performance superiority."**
>
> We want to flag this comment as a potential misunderstanding: these NER papers study neither temporal noise nor time series. Instead, they randomly unlabel named entities in classic NLP datasets. Further, they use short sentences containing 10-20 words. We study time series, which often span hundreds or thousands of timesteps. However, we agree that TENOR is likely applicable broadly and we hope our work inspires such adaptations.
>
> **"Authors should consider more mild assumptions that enable loss correction to be applied to temporal classification."**
>
> We would like to flag another potential misunderstanding: the focus of our work is loss correction for temporal classification. As we mention in Section 3, the conditional independence assumptions we make are common in such sequential classification problems [1-3]. Our findings also apply to our synthetic and real-world datasets, suggesting our assumptions are realistic.
>
> Also as mentioned in Appendix C, our assumptions provide a strong foundation for future work. Existing work in statistical learning theory on highly-dependent sequences may be a promising direction for future research, particularly in the context of temporal label noise, which we define in this work.
>
> **"The proposed TENOR framework is significantly overlapped with the existing method, authors should devise new methods that are specifically designed for temporal classification (orientated to more specific technical challenges)."**
>
> TENOR is designed for our exact setting; it conclusively outperforms existing methods. While all works are inspired by others, TENOR is the first method that addresses our novel problem setting. Beyond TENOR, we formalize a real, unstudied problem, spotlighting a critical weakness of popular methods.
>
> ---
> References
>
> [1] Yoshua Bengio, Réjean Ducharme, and Pascal Vincent. A neural probabilistic language model.
> Advances in Neural Information Processing Systems, 13, 2000.
>
> [2] Yo Joong Choe, Jaehyeok Shin, and Neil Spencer. Probabilistic interpretations of recurrent
> neural networks. Probabilistic Graphical Models, 2017.
>
> [3] Rubanova, Yulia, Ricky TQ Chen, and David K. Duvenaud. "Latent ordinary differential equations for irregularly-sampled time series." Advances in neural information processing systems 32 (2019).

---

> > ### Comment · Reviewer_PKMs · 2023-11-22
> > **Post-rebuttal comment**
> >
> > Dear authors, thank you for your rebuttal, please find my response below:
> >
> > 1. There's no misunderstanding here; I'm not discussing how the noise is generated. A noise could be either randomly or sequentially generated, but the solution in sequential classification should at least utilize sequential information (which is what the papers I mentioned did). Under the context of the existence of sequential correlation, I'm interested in understanding how TENOR excels compared to prior works by utilizing the sequential correlation. Furthermore, there is no necessity in the papers I mentioned that they are specifically designed for short sentences.
> >
> > 2. First of all, I fail to see how conditional independence assumption mentioned in paper [1], I would appreciate if authors can highlight that part. Secondly, if it is true that the conditional independence assumption is true, then what prevents us for using the proof and theoritical results directly from the loss correction paper?
> >
> > 3. Can authors please focus on elaborating the technical novelty in your framework? Yes, I understand that you're **poentially the first** to study a very novel problem, but if it's simply an extension and generalization of existing work to a new setting, that will not be appreciated by the community.
> >
> > Overall, I did not find my concerns are adequately addressed, hence I will adjust my confidence score accordingly.

---

### Official Review · Reviewer_4PX9 · 2023-11-09

**Soundness:** 3 good
**Presentation:** 2 fair
**Contribution:** 3 good
**Rating:** 5
**Confidence:** 4

**Summary:**

The manuscript studies the problem of learning (multi-class) classifiers for sequential data in the presence of temporal label noise, i.e., label flip probabilities (from class i to class j) varies with time. The paper models such temporal label noise via a matrix function (num classes x num classes), as is standard in the literature, but parameterized by time, which is new. The authors propose extensions of the so-called 'forward' and 'backward' loss correction mechanisms for empirical risk minimization on training data with temporal noise, and show that ERM using the proposed losses under noisy labels is equivalent to ERM on the clean data, i.e., their Bayes optimal classifiers coincide. The noise function however also needs to be estimated from the data -- so the authors propose a joint training objective to learn the noise function and the classifier, by parameterizing the noise function via a neural network. Empirical results and various ablations show the effectiveness of the proposed formulation and the learning algorithms.

Overall I think the paper has enough merits but there're some missing details that need to be resolved so that I can better evaluate the contributions. I've some questions below which I'd like the authors respond to in their rebuttal.

**Strengths:**

- Interesting and relevant ML problem and formulation; sequential data arise fairly regularly in many domains, and label noise changing with time also seems well-motivated to warrant a rigorous study.
- Forward and backward sequential loss formulations for temporal noise and theoretical justifications.
- Fairly elaborate and supportive empirical validation, and detailed analysis of various aspects in the proposed approach.

**Weaknesses:**

- Learning a matrix function that changes with time seems very challenging even independently, but learning it together with the classifier is even more so. The paper lacks clarity on why this approach works. If even if there are not any theoretical guarantees for TENOR method, it would be good to discuss why the optimization works and is not derailed by too much label noise (in the experiments, I see even for about 30% noise, the learning is pretty robust).
- More importantly, it's unclear how the TENOR objective imposes regularity across time 't', e.g. if we know the label noise say increases with time in the instances, or varies sinusoidally, etc., it's surprising that the model can learn such Q(t) functions, without any additional regularization or at least a good initial point. The Frobenius norm reg. can't do this. Part of this confusion arises because of my lack of clarity in how Q_w(t) is modeled via a neural network -- are there just one set of weights and an additional 't' parameter (integer) that gives the noise matrix estimates across 't'?
- Theorems 1 and 2 are about equivalences, but there are no forms of estimation/finite sample guarantees in the paper.

**Questions:**

Please respond to the questions in the 'weaknesses' section.

---

> ### Author Response · Authors · 2023-11-17
>
> Thank you for your time and feedback! We respond to your questions below. We hope that this addresses some of the missing details that you were looking for, but please let us know if this is not the case.
>
> **"It would be good to discuss why the optimization works and is not derailed by too much label noise”**
>
> Thanks for your question, we’re glad to have a chance to elaborate!
>
> *"Why the Optimization works”*
>
> A naive approach would be to treat each time-step independently and optimize a separate function or set of weights for each $Q_t$ - this is the VolMinTime baseline in our experiments. The clever part of our implementation is how we link each $Q_t$  using the same function (a neural network with a single set of weights). Now in order to see how we can identify $Q(t)$, we need to consider how the noisy posterior decomposes. Specifically, the noisy-label posterior at time $t$ can be expressed as a product of a noise-transition matrix and a clean-label posterior
>
> $P(\tilde{y}\_t|x\_{1:t}) = Q\_t ^ {\top} P(y\_t|x\_{1:t})$
>
> In our case, $Q_t$ is identifiable provided the posterior distribution is *sufficiently scattered* – i.e., it is not uniformly distributed in the unit-simplex. This follows from an important theoretical result in Non-negative Matrix Factorization (NMF) [1, 2] and motivates a different method to estimate the noise transition-matrix estimation matrix [3], which we use as a baseline in our experiments.
>
> *"Why isn’t TENOR derailed by too much noise"*
>
> If we assume that the posterior distribution is sufficiently scattered, then $Q_t$ is identifiable, and its columns form the minimum-volume simplex that encloses $P(\tilde{y}\_t|x\_{1:t})$. Regardless of the amount of noise, the noisy posterior is a linear combination of the columns of $Q_t$ and the clean posterior. In our experiments, we go up to 40% label noise and find our methods are still effective (see Fig 3 and Appendix E).
>
> In order to efficiently estimate $Q_t$ at all timesteps, we set up the equality-constrained optimization problem in Eq 3 as an augmented Lagrangian objective [4], which is guaranteed to converge under some regularity conditions [4].
>
> **"Theorems 1 and 2 are about equivalences, but there are no forms of estimation/finite sample guarantees in the paper.”**
>
> Thanks for raising this point! We agree that these are infinite sample guarantees. We prioritized these since it seems to be the standard in previous work [5,6]. With that being said, we agree that finite sample guarantees are most important. We will work to see if anything can be said for the finite sample condition. Meanwhile we have updated the text to state this clearly and explained its limitations (Appendix C - Limitations and Future Work).
>
> **“It's unclear how the TENOR objective imposes regularity across time 't… Part of this confusion arises because of my lack of clarity in how $Q(t)$ is modeled via a neural network”**
>
> We briefly discuss how we model $Q_{\omega}(t)$ in Appendix D.2 and are happy to elaborate more here. We use a fully-connected DNN with weights $\omega$ that takes a real-valued time-step as input ($t \rightarrow Q_t$) and outputs a *valid* noise transition matrix as per (Def 1). To ensure that the output is a valid we take the output and:
>
> 1) reshape it into a square matrix $C^2 \times 1$  vector is converted into a (C x C) matrix)
> 2) apply a softmax operation row-wise to ensure entries are valid transition probabilities
> 3) add an identity matrix to ensure diagonal dominance
> 4) and finally we row-normalize to ensure the final matrix is row-stochastic.
>
> Since these operations are differentiable, we can then fit the underlying function $Q(t)$ using standard training algorithms.
>
> In this case, the loss function that we use represents the aggregate volume across all timesteps. The *optimal* weights of the network are therefore the weights that generate the minimum volume transition matrix at each time step. Because the network modelling $Q(t)$ shares one set of weights for all timesteps, we can then model any time-dependent function, due to the universal representation properties of neural networks.
>
> Stepping back, one important point that we want to make is that TENOR can work with any model $Q(t)$ – i.e., beyond the ones that we learn from data using the method described above. For example, we could also use a Neural ODE to parameterize $dQ/dt$ and estimate $Q(t)$ for all t in 1…T by solving an initial value problem. We describe this in greater detail in Appendix C.

---

> ### Author Response · Authors · 2023-11-17
>
> References
>
> [1] Fu, Xiao, et al. "Robust volume minimization-based matrix factorization for remote sensing and document clustering." IEEE Transactions on Signal Processing 64.23 (2016)
>
> [2] Fu, Xiao, Kejun Huang, and Nicholas D. Sidiropoulos. "On identifiability of nonnegative matrix factorization." IEEE Signal Processing Letters 25.3 (2018)
>
> [3] Li, Xuefeng, et al. "Provably end-to-end label-noise learning without anchor points." International conference on machine learning. PMLR, 2021.
>
> [4] Bertsekas, Dimitri P. Constrained optimization and Lagrange multiplier methods. Academic press, 2014.
>
> [5] Patrini, Giorgio, et al. "Making deep neural networks robust to label noise: A loss correction approach." Proceedings of the IEEE conference on computer vision and pattern recognition. 2017.
>
> [6] Kremer, Jan, Fei Sha, and Christian Igel. "Robust active label correction." International conference on artificial intelligence and statistics. PMLR, 2018.

---

### Author Response · Authors · 2023-11-17

We thank all reviewers for their time and feedback!

We are thrilled to see positive reception, and that reviewers recognize the importance of studying temporal label noise, noting that our problem is “interesting and relevant” [4PX9], “non-trivial” [YNZb], “understudied” [PKMs] and “unexplored” [Q9Ck]. We were also glad to see that reviewers recognized our “theoretical justifications” [49X9]  and “extensive experimental results(s)” [Q9Ck] that show “ improvement on multiple datasets against existing state-of-the-art algorithms” [YNZb].

We have addressed questions and comments in responses to each reviewer. To address reviewer feedback and to improve the clarity and broader impact of the work, we have made the following improvements:

1. Improvements to our proposed method yielding more consistent and stable performance, particularly in low-noise settings;

2. Additional experimental results on multiclass classification on both synthetic and real-world data;

3. Experimental results from an additional real-world dataset containing real temporal label noise.

We thank everyone again for their engagement and look forward to answering any remaining questions you may have over the coming days.

---

> ### Author Response · Authors · 2023-11-21
> **Kind reminder of our response**
>
> Dear reviewers,
>
> Thank you again for all of your feedback. We'd like to kindly remind you that we've addressed your concerns in our responses below and we look forward to hearing back. We have worked hard to address your reviews by adding experiments, clarifying some misunderstandings, and reflecting your suggestions in an updated manuscript. As the window for discussion is closing, we want to make sure we have the chance to clarify any lingering concerns.
>
> Thank you once again for your invaluable feedback.

---

### Meta-Review · Area_Chair_FdPV · 2023-12-08

**Metareview:**

This paper focuses on the problem of learning with temporal label noise.

During the rebuttal, reviewers acknowledged that: 1) it contributes to an important problem; 2) some theoretical justifications are provided; 3) the overall written quality is high.

However, during the rebuttal, two important concerns related to theoretical results and technical novelty raised by Reviewer PKMs have not been well addressed.

In the internal discussion, both Reviewer 4PX9 and Reviewer Q9Ck also acknowledged that the concern regarding technical novelty and strong assumptions has not been well addressed.

Overall, the current form cannot be recommended for acceptance. We believe the paper would be strong after addressing the aforementioned concerns. Authors are recommended to revise their paper accordingly.

**Justification For Why Not Higher Score:**

The paper, while addressing an important problem. However, it has not sufficiently differentiated its method from existing ones. Moreover, it clearly explained the novelty of its theoretical results under the class-dependent assumption, where the theoretical results under the class-dependent assumption were already well studied by existing work.

**Justification For Why Not Lower Score:**

NA

---

### Decision · Program_Chairs · 2024-01-16

Reject